# Structural characterization of a highly-potent V3-glycan broadly neutralizing antibody bound to natively-glycosylated HIV-1 envelope

Christopher O. Barnes [1], Harry B. Gristick [1], Natalia T. Freund [2,5], Amelia Escolano [2], Artem Y. Lyubimov [3], Harald Hartweger [2], Anthony P. West Jr.[1], Aina E. Cohen [3], Michel C. Nussenzweig [2,4] & Pamela J. Bjorkman [1]

Broadly neutralizing antibodies (bNAbs) isolated from HIV-1-infected individuals inform HIV-1 vaccine design efforts. Developing bNAbs with increased efficacy requires understanding how antibodies interact with the native oligomannose and complex-type *N*-glycan shield that hides most protein epitopes on HIV-1 envelope (Env). Here we present crystal structures, including a 3.8-Å X-ray free electron laser dataset, of natively glycosylated Env trimers complexed with BG18, the most potent V3/N332$_{gp120}$ glycan-targeting bNAb reported to date. Our structures show conserved contacts mediated by common D gene-encoded residues with the N332$_{gp120}$ glycan and the gp120 GDIR peptide motif, but a distinct Env-binding orientation relative to PGT121/10-1074 bNAbs. BG18's binding orientation provides additional contacts with N392$_{gp120}$ and N386$_{gp120}$ glycans near the V3-loop base and engages protein components of the V1-loop. The BG18-natively-glycosylated Env structures facilitate understanding of bNAb–glycan interactions critical for using V3/N332$_{gp120}$ bNAbs therapeutically and targeting their epitope for immunogen design.

[1] Division of Biology and Biological Engineering, California Institute of Technology, Pasadena, CA 91125, USA. [2] Laboratory of Molecular Immunology, The Rockefeller University, New York, NY 10065, USA. [3] Stanford Synchrotron Radiation Lightsource, 2575 Sand Hill Road, Menlo Park, CA 94025, USA. [4] Howard Hughes Medical Institute, The Rockefeller University, New York, NY 10065, USA. [5] Present address: Department of Clinical Immunology and Microbiology, Sackler Faculty of Medicine, Tel Aviv University, Ramat Aviv, Tel Aviv 6997801, Israel. Correspondence and requests for materials should be addressed to P.J.B. (email: bjorkman@caltech.edu)

In the ongoing fight against the HIV-1 pandemic, the discovery and characterization of broadly neutralizing antibodies (bNAbs) against HIV-1 envelope (Env) fuel new efforts in treatment and management of infection. Next-generation bNAbs protected against and reduced viral loads in humanized mouse[1,2] and non-human primate[3,4] models of infection and exhibited anti-viral activity in human trials[5–9]. Therefore, a vaccine that elicits such antibodies is likely to be efficacious. Despite their promise, unusual properties of HIV-1 bNAbs—such as a high degree of somatic hypermutations[10], long heavy chain complementarity determining region 3 (CDRH3) loops[11], and/or short light chain complementarity determining region 3 (CDRL3) loops[12]—have made it difficult to elicit bNAbs by immunization. In addition, innate features of the viral Env spike that interfere with broad-based immunity include the diversity of Env sequences that arise by mutation[13], a low Env density on the virion surface that interferes with bivalent antibody binding[14,15], and host glycans that shield most of the Env surface[16].

The glycan shield comprises ~50% of the mass of HIV-1 Env, a $(gp120-gp41)_3$ trimer, and consists of N-glycans attached to an average of $30 \pm 3$ potential N-linked glycosylation sites (PNGSs) per gp120-gp41 protomer[17]. The carbohydrates decorating the surface of Env reduce access to protein epitopes and are generally non-immunogenic because they are assembled by host cell machinery[16]. Although under-processed oligomannose N-glycans cover parts of HIV-1 Env such as the $N332_{gp120}$ glycan region of gp120, processed complex-type N-glycans predominate in other regions of Env[18] and protect the CD4 binding site (CD4bs) and the variable loop 3 (V3-loop) of gp120[19]. The production of soluble native-like Env trimers (SOSIPs)[20] has enabled structure determinations of glycosylated Env–bNAb complexes and a better understanding of bNAb epitopes[11,21–24]. However, structural information pertaining to bNAb recognition of highly glycosylated HIV-1 Env trimers in the context of native glycan shields has been difficult to obtain due to chemical and conformational heterogeneity of N-glycans that usually precludes crystallization required for an X-ray structure determination. Thus with one exception[25], all monomeric and trimeric Env crystal structures were solved using glycoproteins produced with exclusively high mannose-only forms of N-glycans, which were usually enzymatically trimmed after being complexed with antibody Fabs[22,26–37]. A single-particle cryo-electron microscopy (cryo-EM) structure of a natively glycosylated HIV-1 Env showed complex-type N-glycans at the gp120-gp41 interface, but many of the remaining glycans were not ordered in the EM map[38,39].

The $V3/N332_{gp120}$ class of HIV-1 bNAbs, exemplified by the PGT121/10-1074 family[40,41], evolved to interact with both glycan and protein components on HIV-1 Env to effect the neutralization of a majority of HIV-1 strains[31,42–44]. These bNAbs possess long CDRH3s that interact with the $N332_{gp120}$ glycan and penetrate the glycan shield to contact the conserved $^{324}GD/NIR^{327}$ peptide motif at the base of the gp120 V3-loop[25,32,42,45]. bNAbs against the $V3/N332_{gp120}$ site isolated from several HIV-1-infected donors can adopt different Env-binding orientations to engage the conserved epitope[29,44] and display a wide array of interactions with surrounding glycans, including glycans at positions $N301_{gp120}$ (the PGT128 bNAb), $N137_{gp120}/N156_{gp120}/N301_{gp120}$ (PGT121/10-1074 family), and $N386_{gp120}/N392_{gp120}$ (PGT135). While under-processed N-glycans presenting as a oligomannose patch predominate in the $V3/N332_{gp120}$ epitope, the recent structure of 10-1074 bound to a natively glycosylated BG505 SOSIP trimer included complex-type N-glycans at positions $N156_{gp120}$ and $N301_{gp120}$[25], and sialic acid-bearing complex-type glycans at the $N156_{gp120}$ position were shown to be critical for maturation of the CAP256 V2 apex bNAb lineage[46]. Thus, providing structural information of bNAbs bound to Env

trimers that include both complex-type and high mannose glycans will facilitate developing strategies for improving bNAb breadth and for design of high-affinity germline-binding immunogens.

BG18 exhibits the highest potency among the $V3/N332_{gp120}$ bNAbs described to date[47]. Isolated from an elite controller infected with clade B HIV-1, BG18 displays a similar breadth of coverage (~64%) across a panel of HIV-1 strains to the PGT121/10-1074 family bNAbs (Supplementary Figure 1), but BG18 neutralizes these strains with a geometric mean $IC_{50}$ value of 0.03 μg/mL, a higher potency than 10-1074 and other bNAbs in human clinical trials[8]. The structure of unliganded BG18 Fab[47] exhibited a cleft between the CDRH2 and CDRH3 loops and a long CDRH3 loop that forms a two-stranded β-sheet as previously observed for PGT121 and 10-1074 Fab structures solved in the absence of HIV-1 Env[40]. However, comparison of the BG18 Fab structure (PDB 5UD9 (https://doi.org/10.2210/pdb5UD9/pdb)) with structures of unbound PGT121 and 10-1074 Fabs (4FQ1 (https://doi.org/10.2210/pdb4FQ1/pdb) and 4FQ2 (https://doi.org/10.2210/pdb4FQ2/pdb)) demonstrated structural differences including: (1) a displaced, shorter, and more compact CDRH3 stabilized by a network of hydrogen bonds, (2) differences in variable light ($V_L$) domain orientation relative to the variable heavy ($V_H$) domain, and (3) a second cleft in the antigen-binding site between CDRH3 and CDRL1/CDRL3. In addition, single-particle electron microscopy showed that BG18 exhibited a different orientation compared with PGT121/10-1074 and other $V3/N332_{gp120}$ bNAbs for binding HIV-1 Env[47].

To better understand the molecular mechanism underlying BG18's interactions with Env glycans and increased potency compared with other $V3/N332_{gp120}$ bNAbs, we determined the crystal structures of natively glycosylated clade A (BG505) and clade B (B41) Envs in complex with BG18 Fab. We used the increased brightness of an X-ray free electron laser[48] (XFEL) to circumvent the limitations of crystal size and improve the resolution to 3.8 Å for our BG18-BG505-35O22 complex. We found that BG18 binds the $V3/N332_{gp120}$ glycan site in a distinct manner relative to PGT121-like bNAbs, primarily through rearrangements in its $V_H$ and $V_L$ domains. Analysis of BG18 interactions bound to a natively glycosylated Env showed engagement with oligomannose glycans near the base of the V3-loop and the conserved GDIR peptide motif. Moreover, BG18's CDRL1 and CDRL2 formed part of a cleft within 8 Å of gp120's variable loop 1 (V1-loop), increasing BG18's protein surface contact with Env. These structures provide valuable information for understanding the promiscuity of $V3/N332_{gp120}$ glycan-directed bNAbs that will facilitate current efforts to evaluate them for therapeutic use in HIV-1-infected humans[8] and to target their epitope for immunogen design[49,50].

## Results

**Structures of natively glycosylated Env-BG18 complexes.** Structural insight into BG18 binding in the context of a natively glycosylated Env was achieved by crystallizing bNAb Fabs complexed with Env trimers expressed and purified from Chinese hamster ovary (CHO) cells[51], which include similar glycoforms as Env trimers expressed in human cell lines[52]. Our crystals comprised natively glycosylated Env SOSIP.664 trimers from clade A (BG505) or clade B (B41) strains bound to Fabs from BG18 and either 35O22, a gp120–gp41 interface-spanning bNAb[53], or IOMA, a VH1-2 CD4bs bNAb[25]. We solved the structures of BG18-BG505-35O22, BG18-B41-35O22, and BG18-BG505-IOMA complexes to resolutions of 4.1, 4.9, and 6.7 Å, respectively (Fig. 1 and Table 1) by molecular replacement (Methods) using synchrotron radiation datasets. Subsequently,

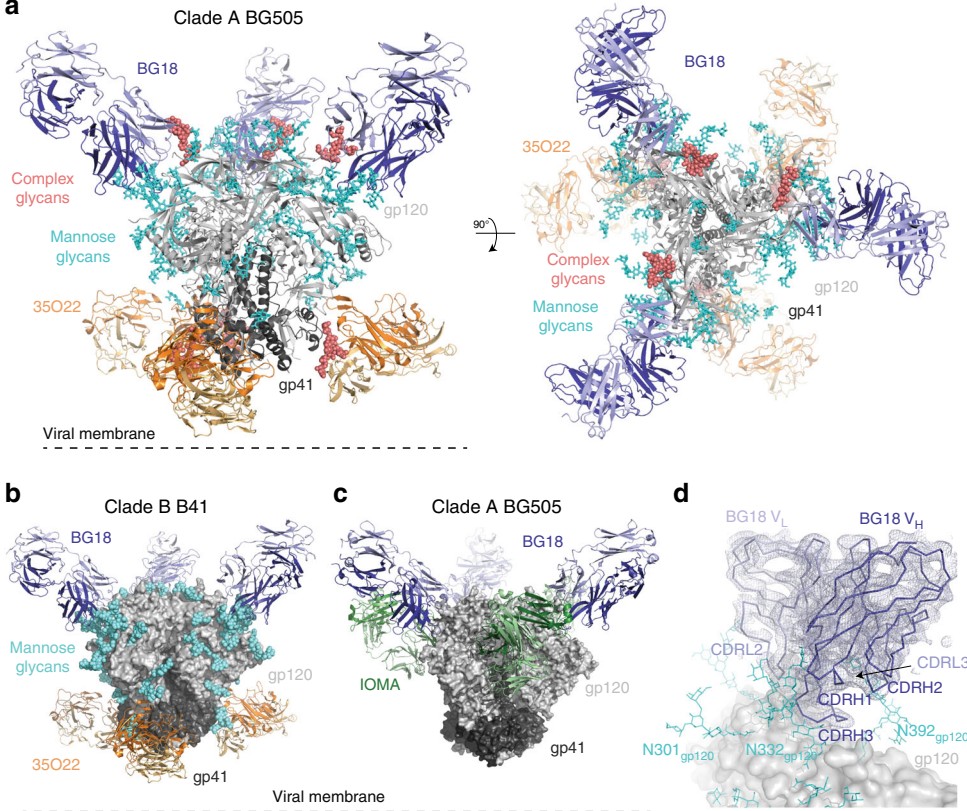

**Fig. 1** Crystal structures of natively glycosylated HIV-1 Env trimers complexed with BG18. **a** Cartoon representation of the 3.8 Å structure of BG505 Env (gp120, light gray; gp41, dark gray) in complex with BG18 (blue) and 35O22 (orange) Fabs. Ordered, native high-mannose glycans (cyan) are represented as sticks, and complex-type glycans (salmon) are shown as spheres. **b** 4.9 Å structure of B41 Env bound to BG18 (blue) and 35O22 (orange). Glycans shown as cyan spheres. **c** 6.7 Å structure of BG505 Env bound to BG18 (blue) and IOMA (green) Fabs. Glycans were not modeled due to the limited resolution. **d** Close-up of the BG18 binding site ($V_H$ in dark blue, $V_L$ in light blue superimposed on the final $2F_o-F_{calc}$ electron density map contoured at 1.2$\sigma$) on the surface of gp120 (gray) from the BG18-BG505-35O22 structure. Ordered glycans (cyan) near the BG18 binding site are represented as sticks

XFEL diffraction data collected by manually targeting individual crystals (~75 μm × 75 μm × 50 μm dimensions) improved intensities of high-angle Bragg reflections (Supplementary Figure 2) as previously observed for XFEL datasets[48,54], resulting in a 3.8 Å resolution structure of the BG18-BG505-35O22 complex from 520 still diffraction images (Fig. 1a). The use of B-factor sharpening[55] in our BG18-BG505-35O222 structures permitted visualization and refinement of most amino acid side chains, particularly at the Fab–Env interface.

The BG18-BG505-35O22, BG18-B41-35O22, and BG18-BG505-IOMA structures each comprised an Env trimer bound to three BG18 Fabs and three 35O22 or IOMA Fabs (Fig. 1a–c), with ordered electron density corresponding to native glycans at the Fab interfaces (Fig. 1d and Supplementary Figure 3a). The orientations of BG18 Fab with respect to gp120 were preserved across the three crystal structures and a low-resolution EM structure of a BG18-BG505 complex[47] (Supplementary Figure 3b–d), indicating that the orientation was conserved across different Env strains and was not an artifact of crystal packing (Supplementary Figure 3e,f). In addition, BG18 did not alter binding modes at the gp41/gp120 interface and CD4bs by 35O22 or IOMA, respectively, as these Fabs adopted similar conformations as previously observed on trimeric Envs[25,27].

Superimposition of the BG18 $V_H$-$V_L$ coordinates in the BG18-BG505-35O22 structure with $V_H$-$V_L$ in unbound BG18 Fab (PDB 5UD9 (https://doi.org/10.2210/pdb5UD9/pdb)) resulted in a 1.3-Å root mean square deviation (rmsd) (240 Cα atoms), demonstrating that BG18 CDR loops (except for CDRL2,

which was disordered in unbound BG18[47]) did not undergo large conformational rearrangements upon binding Env, and thereby maintained the previously observed clefts (Fig. 2a, b). Most notably, the conformation and location of CDRH3 in unbound BG18, which differs from CDRH3s in unbound PGT121 and 10-1074[40,47], were preserved in the Env-bound BG18 structure (Fig. 2c, d). Additionally, interactions with gp120 and glycans at the base of the V3-loop resulted in stabilization of CDRL2.

**BG18 adopts a distinct Env-binding orientation.** A low-resolution Env-bound BG18 structure derived by negative stain EM showed an orientation for BG18 distinct from the orientations of 10-1074 and other V3/N332$_{gp120}$ bNAbs[47]. Despite crystallization contacts involving Fabs (Supplementary Figure 3e, f), BG18 maintained this distinct orientation in our crystal structures compared with orientations in structures of HIV-1 Env trimers with Fabs from 10-1074, PGT122, a PGT121 intermediate, and PGT124[25,27,31,50] (Fig. 3 and Supplementary Figure 4). To analyze these differences, we calculated the rotation and translation for $V_H$-$V_L$ domains of Env-bound BG18 Fab when compared to Env-bound 10-1074 Fab structures. We found that the mature BG18 $V_H$-$V_L$ domains differed by ~78° relative to the orientations of 10-1074 Fab $V_H$-$V_L$ domains (Fig. 3d, e), which contrasts with the ~5° difference between Env-bound PGT122 and 10-1074 Fab orientations. Notwithstanding its different orientation, BG18, 10-1074, and PGT121-like bNAbs share a common mode of interaction with the N332$_{gp120}$ glycan

**Table 1 Data collection and refinement statistics[a,b,c]**

|  | BG18-BG505-35O22 (LCLS: MFX) PDB 6CH7 | BG18-BG505-35O22 (SSRL: 12-2) PDB 6CH8 | BG18-B41-35O22 (SSRL 12-2) PDB 6CH9 | BG18-BG505-IOMA (SSRL 12-2) PDB 6CHB |
|---|---|---|---|---|
| *Data collection* |  |  |  |  |
| Space group | H32 | H32 | H32 | P4$_3$2$_1$2 |
| Cell, Å ($a$, $b$, $c$) | 238.9, 238.9, 354.0 | 239.2, 239.2, 355.3 | 241.6, 241.6, 344.5 | 175.1, 175.1, 454.4 |
| Angles ($\alpha$, $\beta$, $\gamma$) | 90, 90, 120 | 90, 90, 120 | 90, 90, 120 | 90, 90, 90 |
| Wavelength | 1.309 Å | 1.00 Å | 1.00 Å | 1.00 Å |
| Resolution (Å) | 49.66–3.80 | 39.5–4.10 | 39.8–4.95 | 39.81–6.78 |
|  | (3.87–3.80) | (4.32–4.10) | (5.54–4.95) | (7.58–6.78) |
| $R_{pim}$ (%) | – | 7.4 (84.1) | 8.1 (113.7) | 14.2 (139.5) |
| $R_{merge}$ (%) | 53.4 (91.5) | – | – | – |
| I/$\sigma$I | 4.31 (0.32) | 8.2 (1.1) | 5.5 (0.8) | 5.6 (1.2) |
| Completeness (%) | 99.1 (93.8) | 99.8 (99.9) | 99.8 (99.9) | 99.4 (99.6) |
| Multiplicity | 9.2 (3.2) | 11.9 (12.2) | 9.9 (9.8) | 18.5 (19.3) |
| CC$_{1/2}$ | 95.9 (35.4) | 99.7 (61.3) | 99.6 (45.3) | 98.7 (64.8) |
| *Refinement statistics* |  |  |  |  |
| Resolution (Å) | 49.66–3.80 | 39.5–4.10 | 39.8–4.85 | 39.81–6.78 |
|  | (3.87–3.80) | (4.32–4.10) | (5.31–4.85) | (7.58–6.78) |
| Reflections |  |  |  |  |
|   Measured | 354394 | 366818 | 166902 | 238633 |
|   Unique | 38101 | 30813 | 18521 | 13801 |
| $R_{work}$/$R_{free}$ | 24.3/25.7 | 25.3/26.7 | 29.4/31.8 | 37.9/39.1 |
| Wilson B-factor (Å$^2$) | 159.8 | 141.8 | 261.8 | 368.3 |
| Number of atoms |  |  |  |  |
|   Protein | 11,228 | 11,162 | 11,151 | 33,018 |
|   Carbohydrate | 842 | 799 | 779 | 264 |
| R.m.s deviations |  |  |  |  |
|   Bond lengths (Å) | 0.004 | 0.008 | 0.011 | 0.013 |
|   Bond angles (°) | 0.837 | 0.945 | 0.913 | 1.450 |

[a] X-ray free electron data were collected at the Linac Coherent Light Source (LCLS) on Macromolecular Femtosecond Crystallography (MFX) instrument. The XFEL dataset averaged diffraction data from approximately 400 crystals. Conventional synchrotron radiation datasets were collected at the Stanford Synchrotron Radiation Lightsource (SSRL) beamline 12-2, with each dataset generated from a single crystal
[b] Numbers in parentheses correspond to the highest resolution shell
[c] Resolution limits were extended to include weak intensity data[68]

through their CDRH3 loops (Fig. 3c). This interaction includes a structural motif with a consensus R-I-Y-G-V/I-I sequence (BG18 residue numbers 101-106 and 10-1074/PGT122 residue numbers 100-100E; Fig. 2d) encoded by the same antibody D3-3 gene segment, likely explaining the nearly identical N332$_{gp120}$ glycan recognition. Given that CDRH3 is a primary determinant of the interactions of V3/N332$_{gp120}$ bNAbs with Env trimer[29,32,40,44], its displacement in both unbound and Env-bound BG18 (Fig. 2c) rationalizes its orientation of Env binding relative to other V3/N332$_{gp120}$ bNAbs. Consistent with these observations, BG18's footprint at the V3/N332$_{gp120}$ epitope differs from PGT121/10-1074-like bNAbs, such that its interactions with Env glycans and protein components are mediated by different CDR loops relative to the CDR loops used by 10-1074 (Fig. 4a, b and Supplementary Figure 5).

**N-linked glycan interpretation and interactions with BG18.** Modeling of glycans was achieved in our structures by interpreting electron density at PNGSs using $2F_o-F_c$ maps calculated with model phases and in composite annealed omit maps to reduce model bias[56]. In the BG18-BG505-35O22 structures, 17 *N*-linked glycans were modeled into ordered electron density, forming glycan arrays that extended ~22 Å from the Env surface (Fig. 1a and Supplementary Figure 6). *N*-glycans near the BG18 and 35O22 interfaces were modeled as mostly oligomannose (Man$_{5-9}$GlcNAc$_2$) in our structures to avoid over-interpretation of the electron density, except for complex-type glycans at positions N88$_{gp120}$ and N156$_{gp120}$, which were assigned as complex-type based on density for a core fucose in our 3.8 Å XFEL structure (Supplementary Figure 7). Although complex-

type *N*-glycans were modeled at positions N160$_{gp120}$, N276$_{gp120}$, N301$_{gp120}$, and N392$_{gp120}$ in a previous natively glycosylated 10-1074-Env-IOMA structure[25] and observed at positions N160$_{gp120}$, N197$_{gp120}$, and N276$_{gp120}$ by mass spectrometry[18,52,57], our datasets did not show characteristic densities for complex glycans at these positions. Modeling of oligomannose *N*-linked glycans into ordered electron density was also possible in the 4.9 Å BG18-B41-35O22 structure, and for the N332$_{gp120}$ glycan in the 6.7 Å BG18-BG505-IOMA structure (Fig. 1b,c and Supplementary Figure 6). Overlay of BG505 and B41 Env structures showed conservation of the glycan shield surrounding the BG18 binding site (Supplementary Figure 6). Despite resolution limitations that necessitated glycan modeling as predominantly oligomannose at PNGSs shown to attach complex glycoforms, the use of natively glycosylated Env trimers in the crystallization complexes ensured accuracy of bNAb binding orientations and interfaces with Env trimer.

As seen in other PGT121-like bNAb-Env SOSIP structures[25,27,31,50], BG18's primary interaction was with the N332$_{gp120}$ glycan (Man$_9$GlcNAc$_2$), which interfaced with CDRH3, CDRH1, and CDRL2 (Fig. 4a,c; 1170 Å$^2$ total buried surface area (BSA)). This contrasts with PGT121-like bNAb interactions that pack the N332$_{gp120}$ glycan into a groove between CDRH3 and CDRL1/2[25,29,31] (Fig. 4b). Indeed, the observed differences in the orientation of the BG18 Fab light chain on Env influenced the N332$_{gp120}$ glycan conformation, as the glycan D1 arm reached in close proximity to CDRL2 and the gp120 GDIR motif in the BG18-BG505 complex (Fig. 4c), which differs from the interactions of the N332$_{gp120}$ glycan D1 arm with light chain framework region 3 (FWRL3) in the 10-1074-BG505 complex

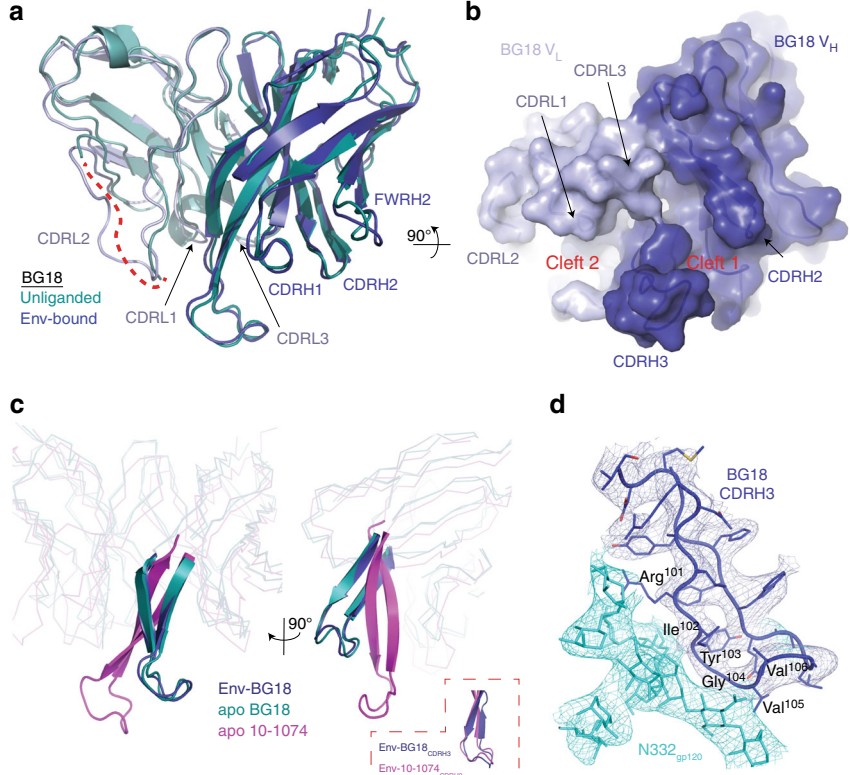

**Fig. 2** Comparison of apo- and Env-bound BG18 variable domains. **a** Superposition of $V_H$-$V_L$ domains (230 Cα atoms) of unliganded BG18 (deep teal; PDB 5UD9 (http://dx.doi.org10.2210/pdb5UD9/pdb)) with BG18 in the BG18-BG505 structure (blue) showed conservation of the BG18 antigen-binding site and ordering of CDRL2 in the BG505-bound structure (dashed red line represents disordered CDRL2 in unliganded BG18). **b** Surface representation of BG18 in the BG505-bound structure showed conservation of BG18 clefts, including the cleft between CDRH3 and CDRL1/3 loops observed in the BG18 apo structure[47]. **c** Ribbon and cartoon representation of the overlay between BG18 (aligned on the Fab $V_H$ domains) in the BG18-BG505-35O22 complex (blue), unliganded BG18 (deep teal), and unliganded 10-1074 (magenta; PDB 4FQ2 (http://dx.doi.org10.2210/pdb4FQ2/pdb)). CDRH3 loops for the three structures are represented as cartoons. Inset: overlay of BG18-BG505 and 10-1074-BG505 (PDB 5T3Z (http://dx.doi.org10.2210/pdb5T3Z/pdb)) CDRH3 loops. **d** Cartoon and stick representation of BG18 CDRH3 recognition of the N332$_{gp120}$ glycan. BG18 residues comprising a conserved structural motif (R-I-Y-G-V/I-I) are labeled. Electron density contoured at 1σ from $2F_{obs}-F_{calc}$ composite annealed omit map calculated with phases from models with the N332$_{gp120}$ glycan and BG18 CDRH3 coordinates omitted to reduce potential phase bias (cyan and blue mesh)

(Fig. 4b). Modeling of Env-bound 10-1074 Fab conformation onto the gp120 subunit from the BG18-BG505-35O22 structure showed clashes between the N332$_{gp120}$ glycan and the CDRL1/2 loops and FWRL3 of 10-1074 (Fig. 4d), suggesting that the distinct BG18 interaction with Env stabilizes a N332$_{gp120}$ glycan conformation that is not possible when binding PGT121/10-1074-like bNAbs[29,31].

In addition to the BG18 interaction with the N332$_{gp120}$ glycan, BG18 makes secondary interactions with the N392$_{gp120}$, N386$_{gp120}$, and N156$_{gp120}$ glycans (Fig. 4a and Supplementary Data 1). These interactions are in contrast with PGT121-like bNAbs that interact with the N137$_{gp120}$, N156$_{gp120}$ glycans in the V1-loop, and N301$_{gp120}$ glycans at the base of the V3-loop (Fig. 4b and Supplementary Data 1)[25,27,32]. When comparing the BG18–Env interaction with the only other structure of a V3/N332$_{gp120}$ bNAb bound to natively glycosylated Env (10-1074–BG505)[25], the rearrangement of BG18's $V_L$ domain reduced BG18's contact with the N301$_{gp120}$ glycan (modeled as a core pentasaccharide in the BG18-BG505-35O22 structure) compared to 10-1074's contacts with the N301$_{gp120}$ glycan (modeled as complex-type biantennary in the 10-1074-BG505-IOMA structure; PDB 5T3X (https://doi.org/10.2210/pdb5T3X/pdb)) (Fig. 4a, b). Interestingly, $V_L$ domain rearrangements allowed BG18 engagement of the N392$_{gp120}$ glycan through its CDR loops (Fig. 4a). The N392$_{gp120}$ glycan threads between the BG18 cleft

that is located between CDRH3 and CDRL1/L3[47] (Fig. 2b), burying ~425 Å$^2$ of Fab surface area against the glycan. This interaction resembles the PGT135 interaction with monomeric gp120 in which ~550 Å$^2$ of gp120 surface area is buried against the N392$_{gp120}$ glycan[44]. However, in contrast to PGT135, which does not neutralize strains lacking the N392$_{gp120}$ or N386$_{gp120}$ glycans, BG18 remains potent against strains lacking glycans at these positions, showing only a ~4-fold reduction in potency (Table 2). Moreover, binding affinities determined by surface plasmon resonance (SPR) showed no effect when the N392$_{gp120}$ glycan was removed from BG505 SOSIP (Supplementary Figure 8). These results are comparable to PGT121-like bNAbs, which equivalently neutralize strains +/− the N392$_{gp120}$ and N386$_{gp120}$ glycans (Table 2), suggesting that BG18 combines binding and neutralization properties of both PGT121-like and PGT135-like bNAbs.

**Molecular details of BG18–Env interactions.** BG18 conserves interactions with the gp120 GDIR peptide motif at the base of the V3-loop that are seen in other V3/N332$_{gp120}$ bNAbs (Fig. 5a, b and Supplementary Figure 5). However, rotation of BG18's light chain relative to other V3/N332$_{gp120}$ bNAbs places only CDRL2 in close proximity to the gp120 GDIR peptide motif, compared to engagement of GDIR by multiple CDR loops and framework

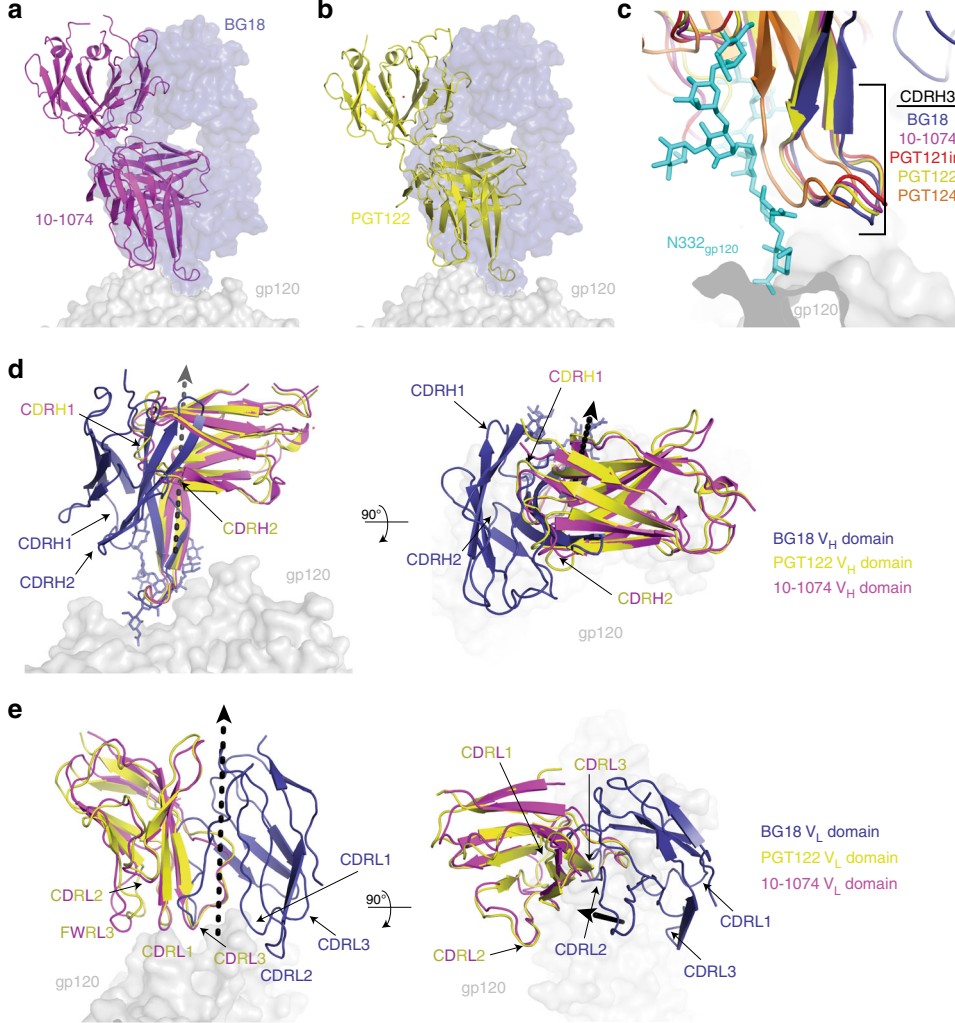

**Fig. 3** Distinct orientation on HIV-1 Env of BG18 compared to other V3/N332$_{gp120}$ bNAbs. BG18 Fab (blue, surface) orientation on gp120 (gray, surface) overlaid with 10-1074 (**a**, magenta; PDB 5T3Z (http://dx.doi.org10.2210/pdb5T3Z/pdb)) and PGT122 (**b**, yellow; 5FYL (http://dx.doi.org10.2210/pdb5T3Z/pdb)) and PGT122 (**b**, yellow; 5FYL (http://dx.doi.org10.2210/pdb5FYL/pdb)) Fabs. **c** Overlay of CDRH3 loop of BG18 (blue), 10-1074 (magenta), PGT122 (yellow), PGT121 precursor (red; PDB 5CEZ (http://dx.doi.org10.2210/pdb5CEZ/pdb)), and PGT124 (orange; PDB 4R2G (http://dx.doi.org10.2210/pdb4R2G/pdb)) after alignment of bound gp120s (gray, surface) demonstrating a conserved structural motif recognizing the N332$_{gp120}$ glycan (cyan). **d**, **e** Comparison of V$_H$-V$_L$ domains of PGT122 (yellow) and 10-1074 (magenta) with BG18 (blue). The V$_H$-V$_L$ domain orientation of BG18 on Env trimer is related by a 78° rotation and 6.8 Å translation about the indicated axis (black dashed arrow) to the 10-1074 variable domains after alignment against gp120 in the BG18-Env and 10-1074-Env structures

regions observed in 10-1074/PGT121-like bNAb recognition (Fig. 5b, c)[25,29,31]. These differences in light chain interactions reflect the germline origins of the BG18 and PGT121/10-1074 light chains, which derive from different VL gene segments (Fig. 5d). However, in an example of convergence toward a common chemical binding mechanism, light chain residues involved in GDIR recognition by BG18 CDRL2 derive from hypermutation from the germline LV3-25*02 gene segment, whereas serines in the CDRL3s of PGT121 and 10-1074 derive from J regions chosen during V–J joining.

B41 Env harbors a GNIR sequence instead of GDIR, allowing us to analyze the D325N$_{gp120}$ substitution in our BG18-B41-35O22 structure. Side chain placement was not possible due to low resolution (4.9 Å), but the BG18 interaction with B41 resembled its interaction with BG505 in the BG18-BG505-35O22 structure (Supplementary Figure 9), suggesting that BG18 recognizes the GNIR motif analogously to how it recognizes GDIR. Consistent with this result, analysis of BG18 neutralization of HIV-1 isolates containing GNIR motifs showed only a 2-fold

loss in potency, by contrast to PGT121 or PGT122, which showed ~6-fold and ~44-fold losses, respectively (Table 2). However, while BG18 S53$_{LC}$ potentially engages N325$_{gp120}$ (Supplementary Figure 8b), the loss of BG18 Q54$_{LC}$ contacts resulted in the disorder of CDRL2 residues 54-60 in our BG18-B41 structure.

Further comparisons between BG18-Env and 10-1074-Env bound structures showed that BG18 CDR loops and light chain framework region 3 are located in close proximity to the gp120 V1-loop (residues 126–158, Supplementary Figure 5), of which residues 140–150 were disordered in the BG18–Env structures (Fig. 6a) and other Env structures[22,25,27,31]. Despite variability in V1 across HIV-1 Envs, we observed engagement of V1-loop protein components with BG18 CDRL2 and FWRL3 residues (Fig. 6b), accommodating the V1-loop into the nearby positively charged cleft formed by CDRH3, CDRL1/L2, and FWRL3 loops (Fig. 6b, c). The electronegativity of this cleft potentially increases protein–protein interactions with BG505 and other HIV-1 strains harboring polar or acidic residues in this region (Fig. 6c). Although PNGSs are commonly found in the V1-loop, the

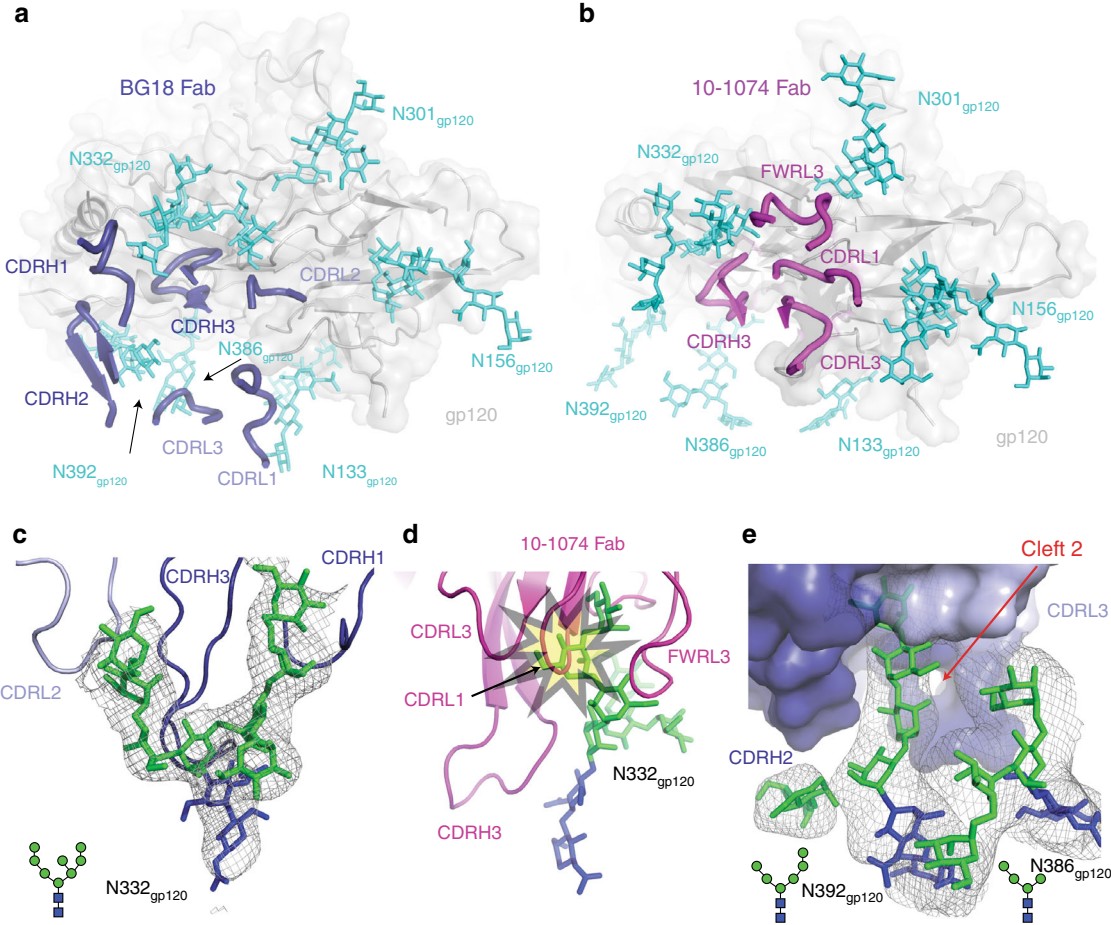

**Fig. 4** Glycan interactions with V3/N332$_{gp120}$ bNAbs in structures including natively glycosylated Env. **a, b** Comparison of the orientations on gp120 (gray surface and cartoon) of the CDR loops from (**a**) BG18 (blue, ribbon) and (**b**) 10-1074 (magenta, ribbon) demonstrating that the BG18 variable domains are rotated clockwise about CDRH3 relative to the 10-1074 variable domains. The distinct BG18 orientation on gp120 resulted in contacts with N156$_{gp120}$, N386$_{gp120}$, and N392$_{gp120}$ glycans (cyan, sticks) in proximity to the V3-base. Red dotted outline: Differences in the N332$_{gp120}$ glycan conformation on BG505 Env bound to BG18 (**a**) or to 10-1074 (**b**). **c** Close-up of BG18 interaction with the N332$_{gp120}$ glycan showing CDRH3, CDRH1 and CDRL2 loops at the glycan interface. **d** Overlay of BG18-BG505 and 10-1074–BG505 (PDB 5T3Z (http://dx.doi.org10.2210/pdb5T3Z/pdb)) structures showed that the BG18-bound N332$_{gp120}$ glycan conformation would clash (yellow star) with light chain CDR loops of 10-1074 and other PGT121-like bNAbs that display nearly identical binding modes. **e** Surface representation of BG18 interactions with the N392$_{gp120}$ and N386$_{gp120}$ glycans showing the N392$_{gp120}$ glycan buried (~800 Å² total BSA) inside cleft 2 located between the CDRH3 and CDRL1/3 loops. The N386$_{gp120}$ glycan associates weakly with light chain BG18 (45 Å² total BSA), but forms branch–branch interactions with the N392$_{gp120}$ glycan. **c, e** Electron density contoured at 1σ from 2$F_{obs}$−$F_{calc}$ composite annealed omit maps calculated with phases from models with glycan coordinates omitted to reduce potential phase bias (gray mesh)

presence of glycans in this region do not notably affect BG18 neutralization (e.g., N133$_{gp120}$, N137$_{gp120}$, and N156$_{gp120}$ glycans; Fig. 6a and Table 2), despite the roof of the cleft being only ~6–8 Å above the V1-loop site. In comparison, the surface of 10-1074 in the comparable region is neutral and does not exhibit a cleft architecture, and only minimally contacts the V1-loop backbone through R94$_{LC}$ (Supplementary Figure 10). Taken together, these observations may explain BG18's increased potency, as BG18 can engage multiple glycan and protein component regions of gp120 (Supplementary Data 1).

## Discussion

Structures of bNAbs complexed with HIV-1 Env trimers have helped elucidate the molecular correlates for anti-HIV-1 antibody breadth and potency. Here we report four crystal structures of the highly potent V3/N332$_{gp120}$ bNAb BG18 bound to natively glycosylated clade A and clade B Envs (Fig. 1a–c). Our structures of clade A BG505 and clade B B41 HIV-1 strains represent only the second example of fully and natively glycosylated Env crystal

structures, with the first being BG505 complexed with another V3/N332$_{gp120}$ bNAb, 10-1074, and with the CD4bs bNAb IOMA[25]. Given the crucial role the Env glycan shield plays in HIV-1's immune evasion strategies[13,58], the prevalence of bNAbs that interact with complex glycans[18,25,58], and the importance of complex glycans in bNAb maturation[46], solving structures containing both complex and high-mannose glycans provides a more complete picture of bNAb recognition of Env epitopes. The newly identified BG18-Env-35O22 crystal lattice system packs solely through Fab interactions (Table 1 and Supplementary Figure 3e), similar to crystals of previously described Env-bNAb complexes[25,27], thus providing an additional system to study HIV-1 Env diversity. Moreover, the improvement in resolution from 4.1 Å using conventional crystals to 3.8 Å resolution by exposing smaller crystal volumes using an XFEL (Supplementary Figure 2) offers the potential to examine natively glycosylated Env trimer–Fab structures to higher resolutions. The demonstration that two HIV-1 Env trimers (BG505 and B41) can be crystallized in different crystal packing lattices without converting their

**Table 2 Glycan and sequence preference for V3/N332gp120 targeting bNAbs in the presence of N332gp120 glycan**

| | IC$_{50}$ values (µg/mL)$^a$ | | | |
|---|---|---|---|---|
| | **BG18** | **10-1074** | **PGT122** | **PGT121** |
| +392 | 0.05 ($n = 67$) | 0.07 ($n = 65$) | 0.21 ($n = 88$) | 0.13 ($n = 224$) |
| −392 | 0.21 ($n = 13$) | 0.08 ($n = 201$) | 0.16 ($n = 16$) | 0.07 ($n = 65$) |
| +386 | 0.05 ($n = 67$) | 0.07 ($n = 234$) | 0.20 ($n = 90$) | 0.12 ($n = 255$) |
| −386 | 0.17 ($n = 13$) | 0.08 ($n = 32$) | 0.20 ($n = 14$) | 0.08 ($n = 34$) |
| +301 | 0.06 ($n = 78$) | 0.07 ($n = 262$) | 0.17 ($n = 99$) | 0.10 ($n = 283$) |
| −301 | 0.60 ($n = 2$)$^b$ | 0.39 ($n = 4$) | 12.30 ($n = 5$) | 12.54 ($n = 6$) |
| +156 | 0.06 ($n = 73$) | 0.07 ($n = 253$) | 0.19 ($n = 97$) | 0.12 ($n = 276$) |
| −156 | 0.05 ($n = 7$) | 0.08 ($n = 13$) | 0.46 ($n = 7$) | 0.11 ($n = 13$) |
| +137 | 0.02 ($n = 9$) | 0.06 ($n = 55$) | 0.10 ($n = 15$) | 0.07 ($n = 60$) |
| −137 | 0.07 ($n = 71$) | 0.08 ($n = 211$) | 0.23 ($n = 89$) | 0.13 ($n = 229$) |
| +133 | 0.03 ($n = 18$) | 0.07 ($n = 65$) | 0.06 ($n = 23$) | 0.10 ($n = 70$) |
| −133 | 0.08 ($n = 62$) | 0.08 ($n = 201$) | 0.28 ($n = 81$) | 0.12 ($n = 219$) |
| D325gp120 | 0.05 ($n = 65$) | 0.05 ($n = 209$) | 0.11 ($n = 89$) | 0.07 ($n = 229$) |
| N325gp120 | 0.10 ($n = 12$) | 0.12 ($n = 47$) | 4.82 ($n = 12$) | 0.41 ($n = 49$) |

$^a$ Geometric mean IC$_{50}$ values were calculated using HIV Antibody Database[74], where $n$ = # of strains. All values were calculated including only strains containing the N332gp120 PNGS
$^b$ Analysis done on the only two strains that met the tested criteria

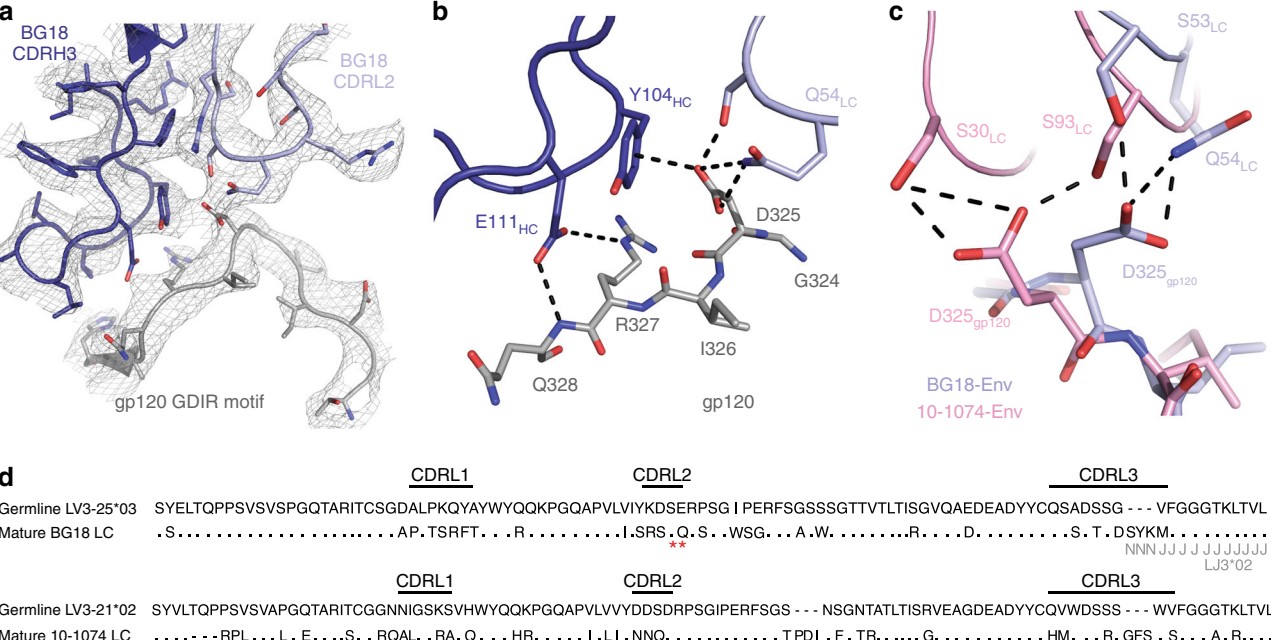

**Fig. 5** Comparison of GDIR recognition by BG18 and 10-1074. **a** Cartoon and stick representation of BG18 CDRH3 (dark blue), BG18 CDL2 (light blue), and gp120 (gray; residues 318-330). Electron density contoured at 1σ from 2F$_{obs}$−F$_{calc}$ final refined electron density map with −100 Å$^2$ B-sharpening is shown (gray mesh). **b** BG18 interactions with the gp120 GDIR motif (G324gp120-D325gp120-I326gp120-R327gp120) at the base of the V3-loop. In common with the PGT121-like bNAbs, CDRH3 E111$_{HC}$ forms a potential salt bridge with R327gp120, while CDRH3 Y104$_{HC}$ stacks against D325gp120. In addition, CDRL2 residues S53$_{LC}$ and Q54$_{LC}$ engage D325gp120 forming potential H-bond interactions. **c** Comparison of BG18 (light blue) and 10-1074 (pink; PDB 5T3Z (http://dx.doi.org10.2210/pdb5T3Z/pdb)) interactions with D325gp120 in the GDIR motif. Engagement of the carboxylate group of D325gp120 is achieved by residues in CDRL2 (BG18) or serine residues in CDRs L1 and L3 (10-1074). Potential H-bonds represented as dashed lines. **d** Alignment of sequences of inferred germline and mature light chains of BG18 and 10-1074. The LV3-21*02 germline V gene segment is used for both 10-1074 and PGT121-like antibodies[40]. Red asterisks indicate residues involved in D325gp120 recognition in the GDIR motif as shown in **c**

glycans to exclusively high-mannose forms provides an impetus for further crystallization efforts using natively glycosylated HIV-1 Envs. Resulting crystal structures can be compared to natively glycosylated Env structures determined by cryo-EM[38], a method that does not require crystallization and can therefore be used for heterogeneously glycosylated samples.

Our structures were consistent with previous evidence that BG18 binds with a distinct orientation compared to the prototype PGT121/10-1074 bNAbs in the V3/N332gp120 glycan-targeting family[47] and showed extensive interactions with both protein and glycan components of gp120 (Fig. 1d). BG18's CDRH3 interactions with GDIR and the N332gp120 glycan are conserved with other V3/N332gp120 bNAbs (Supplementary Figure 5),

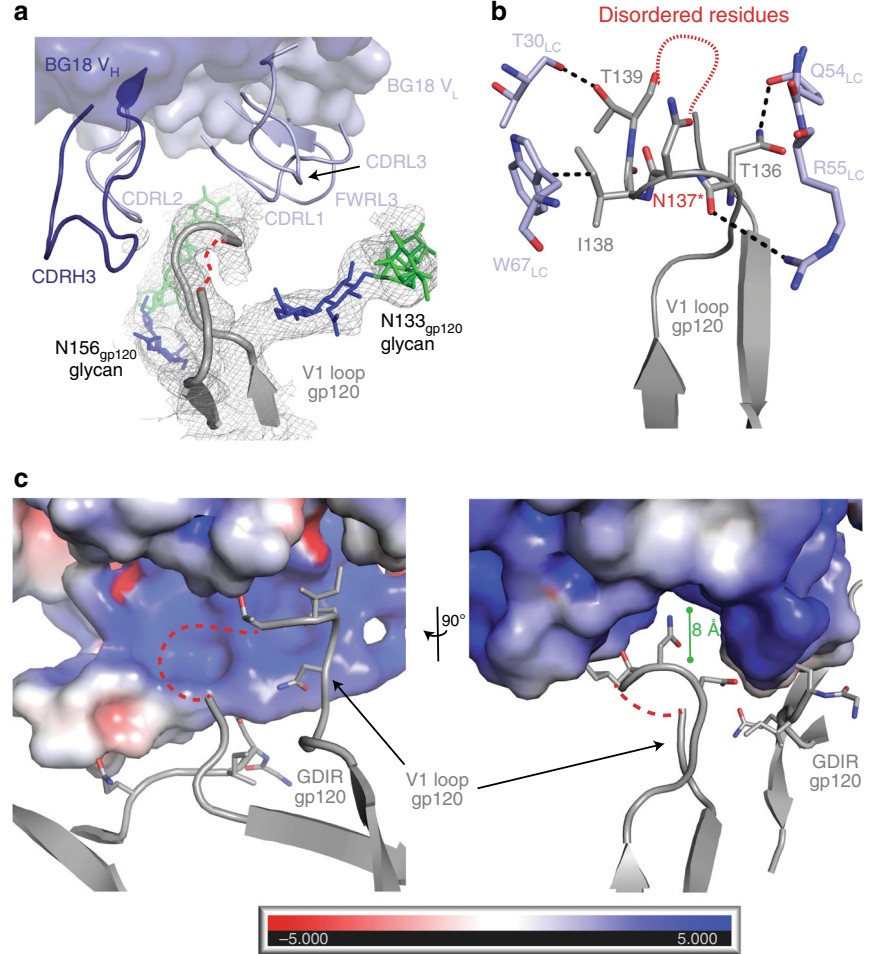

**Fig. 6** BG18 contacts with gp120 V1-loop. **a** Surface and cartoon representation of BG18 $V_H$ (dark blue) and $V_L$ (light blue) loops involved in gp120 V1-loop (gray; residues 128–158, disordered residues depicted as dashed red line) interactions. Electron density contoured at $1\sigma$ from a $2F_{obs}-F_{calc}$ composite annealed omit map is shown (gray mesh). Glycans at position $N133_{gp120}$ and $N156_{gp120}$ are shown. **b** Cartoon and stick representation of residue interactions between BG18 $V_L$ domain (light blue) and the gp120 V1-loop (gray). Potential H-bonding occurs between $T139_{gp120}$ and BG18 $T30_{LC}$ in the CDRL1 loop. In addition, BG18 $W67_{LC}$ in FWRL3 stacks against $I138_{gp120}$. BG18 contacts with gp120 positions 138 and 139 are likely specific to Envs with V1 characteristics similar to BG505, since similar conformations were not observed in our BG18-B41 structure. H-bonds and *pi*-stacking are indicated by black dashed lines. Red asterisk on $N137_{gp120}$ indicates a PNGS. **c** Electrostatic surface potentials with red indicating negative electrostatic potential and blue indicating positive electrostatic potential for BG18 shown with cartoon and stick representation of nearby gp120 elements. BG18 includes a positively charged cleft in the vicinity of the gp120 V1-loop (dashed line indicates disordered region), which may provide increased protein–protein interactions with the gp120 surface in HIV-1 strains harboring charged residues in the V1-loop

serving as the main driver for epitope recognition (Figs 3c and 5b). Strikingly, the D3-3 gene for BG18/10-1074/PGT121 bNAbs that encodes a CDRH3 consensus structural motif responsible for $N332_{gp120}$ glycan interactions plays a role analogous to the VH1-2 gene for VRC01-like bNAbs[24,59,60] for epitope targeting. This common feature, which unites the PGT121 and BG18 bNAb families, suggests that it provides the key interaction in the initial binding of their unmutated ancestor precursors to Env. Interestingly, this common interaction occurs despite different orientations for the rest of the $V_H$-$V_L$ domains (Fig. 4a, b). For example, unlike other V3/N332_{gp120} bNAbs, BG18's light chain CDR loops straddle the V1-loop of gp120 (Fig. 6), increasing surface contact with gp120 protein components. In addition, BG18's interactions with gp120 *N*-glycans differ from the PGT121-like family in both engagement and ability to retain potent neutralization properties. For instance, despite extensive interactions with glycans at positions $N392_{gp120}$ and $N386_{gp120}$ (Fig. 4a, e and Supplementary Data 1), BG18 can potently neutralize HIV-1 strains lacking these glycans, by contrast to the

more weakly neutralizing V3/N332_{gp120} bNAb, PGT135, which relies on $N332_{gp120}$, $N392_{gp120}$, and $N386_{gp120}$ glycans for its anti-HIV-1 activity[44] (Table 2). Consistent with this observation, analysis of neutralization by BG18 and PGT121/10-1074 family bNAbs of HIV-1 strains including the $N332_{gp120}$ glycan, but with and without glycans known to interact with bNAbs targeting the V3/N332_{gp120} epitope, showed that BG18 and 10-1074 retained their anti-HIV-1 potency (mean $IC_{50}s < 0.6 \mu g$/mL), whereas PGT121/PGT122 potency was abrogated upon removal of the $N301_{gp120}$ glycan (Table 2). Furthermore, PGT122 showed sensitivity to $D325N_{gp120}$ mutations in the GDIR peptide motif. Taken together, these differences illustrate the divergence of solutions evolved by bNAbs to target this epitope, and BG18's successful strategies to accommodate Env sequence diversity.

Recent studies showed that priming with designed SOSIP Env trimers that bind weakly to the common inferred germline sequence (iGL) of PGT121 and 10-1074[50] could elicit bNAbs resembling PGT121 in PGT121 iGL Ig knock-in mice[49].

However, the iGL of BG18 did not bind detectably to the tightest-binding designed priming immunogen 11MUT$_B$ (Supplementary Figure 11a), which can be rationalized by one of several differences in BG18 and PGT121/10-1074 recognition of Env. First, mature BG18 CDRL2 residues are involved in GDIR recognition with S53$_{LC}$ and Q54$_{LC}$ engaging D325$_{gp120}$ of GDIR (Fig. 5b). Comparison of iGL and mature BG18 CDRL2 amino acid sequences showed that four of five residues were mutated ($^{50}$YKDSE$^{54}$ vs. $^{50}$SRSSQ$^{54}$, respectively) (Fig. 5d). Not only does the germline CDRL2 have increased bulk, but also acidic residues flank S53$_{LC}$, which interacts with D325$_{gp120}$. In contrast, iGL sequences for CDRL1/3 in PGT121/10-1074 show conservation of serines responsible for interacting with D325$_{gp120}$ in GDIR and flanking residues (Fig. 5d). Thus, since the 11MUT$_B$ priming immunogen contains no substitutions in GDIR, BG18 iGL likely shows a reduced ability to interact with this region compared with PGT121/10-1074 iGLs. An additional predicted impediment to iGL BG18 binding to 11MUT$_B$ is the relative proximity of light chain CDRs to the gp120 V1-loop (Fig. 6). Since the CDRs are the most heavily substituted between iGL and mature sequences, altering the chemical properties of this cleft could negatively affect binding. Moreover, 11MUT$_B$ harbors seven mutations in the V1-loop necessary for iGL PGT121 binding[50], which likely clash with iGL BG18 given its V1-loop interactions (Supplementary Figure 11b). Finally, mature BG18 is heavily substituted compared to iGL sequences (35 and 26 heavy chain and light chain amino acid mutations, respectively). Previous structural studies of a PGT121 intermediate bound to BG505 Env showed that orientations are defined early during maturation and that differences in amino acid composition can alter antibody footprints on gp120[31]. Thus, it is possible that the conformation of iGL BG18 is incompatible with 11MUT$_B$ binding.

Although eliciting BG18-like bNAbs would require a different set of designed immunogens than being used for PGT121/10-1074 bNAbs[49,50], they might be easier to elicit because of a shorter CDRH3 than PGT121/10-1074 bNAbs. By elucidating the molecular details of BG18's distinct interaction with the V3/N332$_{gp120}$ epitope, the structural information reported here will facilitate future immunogen design efforts.

## Methods

**Protein expression and purification**. Fabs from BG18 (including a N26Q$_{HC}$ substitution[47]), 35O22, and IOMA IgGs were produced as previously described[37]. Briefly, Fabs were expressed by transiently transfecting HEK293-6E cells (National Research Council of Canada) with vectors encoding the appropriate light chain and C-terminal 6x-His tagged heavy chain genes. Secreted Fabs were purified from cell supernatants using Ni$^{2+}$-NTA affinity chromatography (GE Healthcare), followed by size exclusion chromatography (SEC) with a Superdex200 16/60 column (GE Healthcare). Purified Fabs were concentrated and maintained at 4 °C in storage buffer (20 mM Tris pH 8.0, 150 mM NaCl, and 0.02% sodium azide).

Genes encoding BG505 SOSIP.664 gp140[20] or B41 SOSIP.664 gp140[61] trimers were stably expressed in CHO Flp-In$^{TM}$ cells (Invitrogen) as described[51] using cell lines kindly provided by John Moore (Weill Cornell Medical College). Plasmids encoding the BG505 SOSIP.664 gp140 N392 gene variant (see Supplementary Table 1 for primer sequences) was transiently expressed in HEK293-6E cells (National Research Council of Canada) as previously described[22]. In both cases, secreted SOSIP trimers expressed in the absence of kifunensine were isolated from cellular supernatants using 2G12 immunoaffinity chromatography by covalently coupling 2G12 IgG monomer to an activated-NHS Sepharaose column (GE Healthcare). Trimers were eluted using 3 M MgCl$_2$ and immediately dialyzed into storage buffer before SEC purification with a Superdex 200 16/60 column (GE Healthcare) against the same buffer. Peak fractions pertaining to SOSIP trimers were pooled and repurified over the same column and buffer conditions. Twelve 1.0-mL fractions were stored separately at 4 °C.

**Crystallization of BG18–Env complexes**. Complexes for crystallization were produced by an overnight incubation of SOSIP with BG18 and 35O22 or IOMA Fabs at a 1:1:1 molar ratio, and subsequently concentrated to 5–8 mg/mL by centrifugation with a 30-kDa concentrator (Amicon). Initial matrix crystallization trials were performed at room temperature using the sitting drop vapor diffusion

method by mixing equal volumes of protein sample and reservoir using a TTP LabTech Mosquito robot and commercially available screens (Hampton Research and Qiagen). Initial hits were optimized and crystals were obtained for BG18-BG505-35O22 and BG18-B41-35O22 in 0.1 M Tris pH 8.0, 5% Tacsimate pH 8.0, and 14% polyethylene glycol (PEG) 3350 at 20 °C. BG18-BG505-IOMA crystals were obtained in 0.1 M citric acid pH 3.7, 16% PEG 3350 at 20 °C. Crystals were cryo-protected stepwise with reservoir and a final 20% v/v glycerol concentration before being cryopreserved in liquid nitrogen.

**Structure determination and refinement**. Conventional X-ray diffraction data were collected for BG18-Env complexes at the Stanford Synchroton Radiation Lightsource (SSRL) beamline 12-2 on a Pilatus 6M pixel detector (Dectris). Data from a single crystal for each complex were indexed and integrated in XDS[62], and merged with AIMLESS in the CCP4 software suite[63]. Structures were determined by molecular replacement in PHASER[64] using a single search with coordinates of an a glycosylated gp120-4 protomer (PDB 5T3Z (https://doi.org/10.2210/pdb5T3Z/pdb)), BG18 Fab (PDB 5UD9 (https://doi.org/10.2210/pdb5UD9/pdb)), and 35O22 Fab (PDB 4TOY (https://doi.org/10.2210/pdb4TOY/pdb)) or IOMA (PDB 5T3Z (https://doi.org/10.2210/pdb5T3Z/pdb)). Models were refined using B-factor refinement in CNS[65] and Phenix[56], followed by several cycles of manual building with B factor sharpening in Coot[55,66]. Glycans were initially interpreted and modeled using $F_o - F_c$ maps calculated with model phases contoured at 2σ, followed by 2$F_o - F_c$ simulated annealing composite omit maps in which modeled glycans were omitted to remove model bias[56]. N-linked glycans identified at individual PNGSs in our crystallographic studies on both BG505.664 and B41.664 SOSIP trimers were generally consistent with the mixture of glycans observed by mass spectroscopy[18,57] and previous crystallographic studies of a natively and fully glycosylated Env trimer[25]. Therefore, modeling of complex-type glycans at positions N88gp120 and N156gp120 was primarily determined by the presence of electron density characteristic of a core fucose, which when modeled, slightly lowered $R_{free}$ values. Additional details of glycan modeling and coordinate refinement were followed as previously described[25,67]. Inclusion of higher resolution data with weak intensities improved refinement behavior and stereochemistry as described[68].

XFEL diffraction experiments were performed at the MFX endstation of LCLS using a standard goniometer setup[48], 9.5 keV X-ray pulses with 40 fs duration and a 5-μm beam focus at the crystal interaction point. Diffraction images were recorded on a Rayonix MX325 detector data and were integrated with IOTA[69] using the data reduction algorithms implemented in cctbx.xfel[70]. Of the 627 collected images, 589 contained discernible diffraction; of these, 570 were successfully integrated and yielded correct crystal symmetry and unit cell values. Scaling, post-refinement, and merging was carried out using PRIME[71]. Of the 570 integrated diffraction images, 526 were included in the final merged dataset, which was complete (99.1%) to 3.8 Å, and exhibited good multiplicity (9.2-fold) and reasonable merging statistics (Table 1). Phases were generated by molecular replacement, using our refined 4.1 Å synchrotron structure with glycans omitted as a search model.

Superposition and figures were rendered using PyMOL (Version 1.5.0.4 Schrodinger, LLC), and protein electrostatic calculations were achieved using APBS and PDB2PQR webservers[72]. BSAs were determined with PDBePISA using a 1.4-Å probe[73]. Potential hydrogen bonds were assigned using a distance of <3.6 Å and an A-D-H angle of >90°, while the maximum distance allowed for a van der Waals interaction was 4.0 Å. Putative H-bonds, van der Waals assignments, and total BSA should be considered tentative, owing to the relatively low structure resolutions.

**Binding experiments**. SPR experiments were carried out on a Biacore T100 (Biacore) using a standard single-cycle kinetics method as previously described[22,37]. Briefly, a CM5 chip, primarily amine coupled with Protein A, was used to immobilize 8ANC195 IgG, a gp120-gp41 interface bNAb[37]. Remaining Protein A sites were blocked with 1 μM human Fc. BG505 HIV-1 SOSIP trimers were captured by injecting 10 μM solutions at a flow rate of 30 μL/s for 180 s. Mature BG18 Fab was injected at decreasing concentrations (3-fold dilution series with a starting top concentration of 110 nM) at 30 μL/s for 60 s and allowed to dissociated for 300 s. Kinetic analyses were done after subtraction of reference curves to obtain $k_a$, $k_d$, and $K_D$ values for a 1:1 binding model with or without a bulk refractive index change correction as appropriate (Biacore T200 Evaluation software).

An ELISA to evaluate binding of mature and iGL BG18 IgG to 11MUT$_B$ SOSIP[50] was performed by coating of High-Bind 96-well plates (Corning #9018) with 50 μL per well of a 2-μg/mL solution of purified 11MUT$_B$ in PBS overnight at 4 °C. Plates were washed six times with washing buffer (1× PBS with 0.05% Tween 20 (Sigma-Aldrich)) and incubated in blocking buffer (1× PBS with 1% non-fat milk) for 1 h at room temperature (RT). Immediately after blocking, IgGs were added in blocking buffer and incubated for 2 h at RT. Antibodies were assayed at a 5-μg/mL starting dilution and seven additional 3-fold serial dilutions. Plates were washed six times with washing buffer and then incubated with anti-human IgG secondary antibody conjugated to horseradish peroxidase (HRP) (Jackson Laboratories) in washing buffer at a 1:5000 dilution. Plates were developed by addition of the HRP substrate, ABTS (Life Technologies), and absorbance was measured at 405 nm with an ELISA microplate reader (FluoStar Omega, BMG Labtech).

**Data availability**. Coordinates and structure factors reported in this manuscript have been deposited in the Protein Data Bank with accession codes 6CH7, 6CH8, 6CH9, and 6CHB. Other data are available from the corresponding author upon reasonable request.

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

## Acknowledgements

We thank J. Vielmetter, S. Khomandiak, and the Caltech Protein Expression Center for producing proteins, J. Kaiser and the beamline staff at SSRL for data collection assistance, and members of the Bjorkman and Nussenzweig laboratories for critical reading of the manuscript. We also thank Silvia Russi, Clyde Smith, Jinhu Song, and Alexander Batyuk for user support during our MFX experiment and the lab of Guillermo Calero (University of Pittsburgh) for providing loop supports to mount multiple crystals during XFEL experiments. C.O.B holds a Hanna Gray Fellowship from the Howard Hughes Medical Institute and a Postdoctoral Enrichment Program Award from the Burroughs Wellcome Fund. This research was supported by the National Institute Of Allergy And Infectious Diseases of the National Institutes of Health Grant HIVRAD P01 AI100148 (P.J.B.); (the content is solely the responsibility of the authors and does not necessarily represent the official views of the National Institutes of Health), the Bill and Melinda Gates Foundation (Collaboration for AIDS Vaccine Discovery Grant OPP1124068 (M.C.N./P.J.B.)), and the Molecular Observatory at Caltech supported by the Gordon and Betty Moore Foundation. Use of the Linac Coherent Light Source (LCLS) and use of the Stanford Synchrotron Radiation Lightsource (SSRL), SLAC National Accelerator Laboratory, is supported by the U.S. Department of Energy, Office of Science, Office of Basic Energy Sciences under Contract No. DE-AC02-76SF00515. The SSRL Structural Molecular Biology Program is supported by the US Department of Energy Office of Biological and Environmental Research, and by the National Institutes of Health (NIH), National Institute of General Medical Sciences (including P41GM103393).

## Author contributions

C.O.B., A.P.W. Jr., M.C.N., and P.J.B. conceived the experiments; C.O.B. optimized the crystallography conditions, solved and analyzed the crystal structures; C.O.B., H.B.G., and A.P.W. Jr. performed computational and structural analyses of BG18-class and PGT121/10-1074-class antibodies; C.O.B. and H.B.G. purified the proteins, performed and analyzed SPR binding experiments; C.O.B., A.Y.L., and A.E.C. performed the XFEL data collection and analyses; N.T.F. was responsible for isolating BG18 monoclonal antibodies; A.E. and H.H. performed ELISA experiments; C.O.B., A.P.W., M.C.N., and P.J.B. wrote the paper, on which all principal investigators and authors commented.

## Additional information

**Competing interests:** The authors declare no competing interests.

