## [Peer Review File(PDF 857 kb) · Nature Communications]

Reviewers' comments:

Reviewer #1 (Remarks to the Author):

The exact delineation of broadly neutralizing (bNAb) targets on the HIV-1 spike fuels vaccine design efforts. One bNAb target is the V3-base and surrounding glycans. This group of authors recently isolated the the most potent against this epitope cluster, BG18. In the current paper the authors describe the high-resolution structure of this bNAb in complex with the BG505 (clade A) Env trimer. In addition, the authors provide a lower resolution structure with the B41 (clade B) Env trimer. These structures reveal the molecular footprint on the Env trimer, which involves the GDIR motif at the V3-base, glycans at positions N332, N386 and N392, and residues in the V1-loop (138, 139). Furthermore, BG18 has a different angle of approach compared to other V3-base targeting bNAb such as PGT121 and family members. BG18 has some similarities with PGT135 such as contacts with the glycans at N386 and N392. This is an excellent study, and I have a few comments that can strengthen the manuscript.

The conclusions of the study, including the definition of critical contact residues, often rely only of structural information, and/or on sequence comparison of neutralization-sensitive and -resistant primary HIV-1 viruses. Although the latter analysis provides nice support for the structural information, it would be good to provide other supporting evidence by binding analyses such as SPR and/or mutant viruses. The authors have done so for the N392 glycan but it would be good to see similar analyses for the GDIR vs GNIR motifs (see also below), for the N386 glycan and for the V1-residues 138 and 139.

The comparison of BG18 with other V3-base bNAbs could be extended a bit. The comparison between BG18 and 10-1074 in Fig. 5 is great, but why not broaden the comparison on several parameters. The different angles of approach of various V3-base bNAbs are already compared in Fig. S4 and possibly this figure could be extended.

1. The authors could include a neutralization potency/breadth plot for PGT121, PGT128, PGT124, 10-1074, PGT135 and BG18 showing that BG18 is the most potent (assuming the relevant data is available in the existing literature).
2. The protein contacts (buried surface area per amino acid) could be tabulated for each antibody. These would include the amino acid contacts of the GDIR motif, in the V1-loop, and others if relevant.
3. The glycan contacts (buried surface area per individual glycan) could be tabulated. I assume this would include the glycans at N133, N137, N156, N301, N332, N386, N392, and others if relevant.
4. The (overlapping) footprints of the different bNAbs could be shown on the trimer structure.

Despite the low resolution, the B41 structure might be exploited a bit more. The authors observe the difference between the GDIR and GNIR motifs. It would be good to see SPR data on the B41 trimer containing a GDIR motif and the BG505 trimer with a GNIR motif to show the effect of the amino acid substitution on binding affinity.

If I got this correct, by default, glycans were modeled as Man5-9GlcNAc2 glycans, UNLESS density was clearly interpretable for fucose and/or moieties emanating from the oligosaccharide core. However, there is now a wealth of emerging site-specific glycan composition data from the Paulson lab and in particular the Crispin lab that be exploited here. Instead of modeling Man5-9GlcNAc2 glycans by default, the authors could model the most commonly found glycoform species from the site-specific analysis of the Crispin lab (unless density is clearly interpretable and indicates that a different glycoform is present compared to the most common one in the Crispin analysis).

Some glycans are mentioned as part of the V3-base (N386, N392 in abstract and end of introduction;

and also "N137, N156 and N301 at the base of V3", results page 10). This is ofcourse correct for N301, but many virologists would not consider N386 and N392, and N137 and N156 as being located at the V3 base, but in V4 or V1, respectively. I realize that they are close to the V3-base in the 3D structure, but the wording is a bit ambiguous. I would propose to change the wording to express that they are in close proximity of the V3-base.

The designation "V3-targeting bNAbs" (title and elsewhere) might cause some confusion. Traditionally, V3 epitopes have been regarded as targets for nonneutralizing Abs or NAbs for only unusually sensitive viruses that expose their V3 constitutively. Such Abs usually target the tip of V3. In contrast, bNAbs such as BG18 target the V3-base and surrounding glycans (and even some residues in V1). Rather than V3 antibodies I would suggest terming them V3-base or V3-glycan antibodies.

Although the structures contain the IOMA and 35O22 bNAbs as well as BG18, they are not further discussed in the paper. I assume that the structures did not reveal anything new on these bNAbs, but if that is the case, perhaps the authors could acknowledge this in one sentence.

In the first paragraph of the discussion the authors mention that their new structure represents the second structure with fully and naturally glycosylated Env. I assume this is true for the existing crystal structures, but what about the existing cryo-EM structures? Perhaps that authors can explain/expand.

Table 2. What does the * stand for?

Reviewer #2 (Remarks to the Author):

Barnes et al. report three crystal structures of a potent V3/N332-directed broadly neutralizing antibody (bNAb) BG18 in complex with natively-glycosylated, clade A and clade B SOSIP Env trimers at 3.8Å, 4.9Å and 6.7Å, respectively. A 3.8 Å X-ray free electron laser dataset was necessary to give an electron density map that allowed interpretation of most amino acid side chains, in particular, those at the antibody-Env interface. These structures show a different angle of approach by BG18 to gp120 when compared to other members of the PGT121/10-1074 bNAb family. This unique orientation allowed additional contacts with glycans at N392 and N386 near the V3 base, as well as contacts with the polypeptide chain of the V1, which might explain the increased potency of BG18 in neutralization. The structural information could potentially guide development of V3/N332-directed therapeutic antibodies and immunogen design targeting this important epitope.

The manuscript is clearly written, and data are solid and well presented. It should be of interest to the HIV vaccine community and general readers of Nature Communications. A small concern is their interpretation of glycan density at relatively low resolutions. This is a very difficult problem because of the intrinsic heterogeneity of glycosylation and the authors have indeed tried hard to "avoid over-interpretation of the electron density". The question is – how reliable is it to build complex-type glycans simply based on a core fucose? Although core-fucosylation is mainly associated with complex sugars, it has been reported that short high-mannose sugars can also be core-fucosylated. In addition, as the authors pointed out, it seems that some complex type N-glycans previously modeled at positions N160, N276, N301 and N392 are not supported by their current data sets. It is therefore generally a good practice to interpret only what the map can show. Otherwise, it would be misleading, particularly for those non-structural biologists who would believe everything built in the model.

Reviewer #3 (Remarks to the Author):

This MS reports crystal structures of two different SOSIP constructs (a clade A, BG505) and a clade B (B41) bound with the Fab of a recently characterized, V3-directed antibody, called BG18. The structures include BG18 with BG505 and a gp120/gp41 interface Fab (35O22), BG18 with BG505 and IOMA (a CD4bs Fab), and BG18 with B41 and 35O22. The gp140s have native glycosylation, which is relevant to this class of antibodies, which contact both protein and glycan. A data set for the first of these three structures, recorded on the X-FEL at LCLS, extended the resolution of the structure modestly but usefully (from 4.1 Å to 3.8 Å).

Analysis of BG18 bound with gp140 adds to the increasingly extensive portfolio of structures of various bnAbs in a more or less "functional" context. The structure determinations are solid, the descriptions detailed (too detailed for this reader), and the comparisons with PG121 and other V3 antibodies useful. With revision, the MS will be appropriate for publication by Nature Communications.

What do I mean by "with revision"? Four principal points.

(1) The description of the contacts, comparisons, etc., is unnecessarily complicated. As far as I could tell, what matters is that the footprint of this antibody is somewhat different from the footprint of the others, in defined ways (with consequences for the overall pose of the domains) that should be relatively easy to outline. That is all a reader can comprehend, without having the coordinates, and when one has the coordinates, a brief statement of what to look for is all that's needed.

(2) The authors try to make a case for the advantages of this particular antibody (with respect to other V3 antibodies) -- as a therapeutic, or perhaps as the sort of V3 antibody one might like to elicit with a suitably designed immunogen. The tone throughout smacks a bit of "my daddy can beat up your daddy" -- methinks the authors do protest too much, however, so I come out a skeptic instead. A succinct comparison of pros and cons in the Discussion is fine, but otherwise, just give the readers the facts and let them come to their own conclusions.

(3) The X-FEL data were clearly helpful. But the comparison isn't a fair one, as the characteristics of the x-ray beam they used at SSRL were of course different, not just in brightness but also in collimation, diameter, etc. In other words, there are advantages to a very small, well-collimated, very bright beam if your crystal is small (these weren't SO small), but perhaps GMCA-CAT at APS would have done as well (with a 10 micron beam, helical scanning or similar strategy, etc.) A comparison isn't the purpose of this report, of course, so I believe that the authors should remove "as revealed by an X-ray free electron laser" from the title. Yes, given the resources these authors had to hand, that's how they did it, and they obviously did a great job. But the hidden implication is that you need an X-FEL rather than a better synchrotron beam. This reviewer is unconvinced.

(4) "Angle of approach" is a poor term (pace Andrew Ward). For several reasons. First, there's a good, much better, term used generally in chemical biology -- "pose". (Note: only four letters!). Second, what really matters in the case of antibodies is the footprint and its details, not the "angle" (badly defined in any case, as "angle" is a 2D term; in 3D, there are at minimum two parameters, e.g., direction cosines of an axis, and one needs a common reference axis in the two structures as well).

There is also a point that I'd like to see addressed, although the resolution of one of the structures limits what what one can say. Nonetheless, I'd like to know whether the IOMA bound gp140 is in any way different in conformation from the 35O22 bound gp140 (BG505 in both cases). The BG12 Fab may have an influence, potentially minimizing any conformational difference induced by IOMA, but a comment would be useful.

A few trivial items:

lines 30-32 Our structures show that common, D-gene encoded residues have conserved contacts with the N332 glycan and with the GDIR peptide motif, even though the pose of the Fab is different from those of V3/N332 bnAbs in the PGT121/10-1074 family. (or, to save words, "from those of PGT121/10-1074-family V3/N332 bnAbs"). [I believe we should reserve "reveal" for religious revelation; "show" is quite good enough for down-to-earth science. I concede that "reveal" is OK in sentences like "Einstein's theories revealed a deep connection between space and time", but that was a general, epistemological revelation, not a simple statement of observation. Note also active voice in subordinate clause.]

line 198 contrasts with interactions (contrasts doesn't take a definite object)

lines 206-207 N137 and N156 are NOT at the base of the V3 loop -- reword the sentence

line 232 contrasts with

line 236 misuse of "rationalize that". These differences in light chain interactions reflect the germline origins of the BG18 and PGTxxx light chains, which derive from different VL gene segments.

line 241 ...GNIR sequence instead of GDIR, allowing us to analyze the D325N substitution... (current version ungrammatical)

line 245 ...GNIR motif analogously to how it recognizes GDIR.

lines 254 and 260 delete "surprisingly" Let the reader decide whether it is surprising or not.

We thank the reviewers for their insightful comments and suggestions regarding our manuscript, **“Structural characterization of a highly-potent V3-glycan broadly neutralizing antibody bound to natively-glycosylated HIV-1 Envelope.”** Based on these comments, we have prepared a revised manuscript and include point-by-point responses below. We would be very grateful for a rapid decision, as this is a very competitive area of research.

Reviewer #1 (Remarks to the Author):

The exact delineation of broadly neutralizing (bNAb) targets on the HIV-1 spike fuels vaccine design efforts. One bNAb target is the V3-base and surrounding glycans. This group of authors recently isolated the the most potent against this epitope cluster, BG18. In the current paper the authors describe the high-resolution structure of this bNAb in complex with the BG505 (clade A) Env trimer. In addition, the authors provide a lower resolution structure with the B41 (clade B) Env trimer. These structures reveal the molecular footprint on the Env trimer, which involves the GDIR motif at the V3-base, glycans at positions N332, N386 and N392, and residues in the V1-loop (138, 139). Furthermore, BG18 has a different angle of approach compared to other V3-base targeting bNAb such as PGT121 and family members. BG18 has some similarities with PGT135 such as contacts with the glycans at N386 and N392. This is an excellent study, and I have a few comments that can strengthen the manuscript.

The conclusions of the study, including the definition of critical contact residues, often rely only of structural information, and/or on sequence comparison of neutralization-sensitive and –resistant primary HIV-1 viruses. Although the latter analysis provides nice support for the structural information, it would be good to provide other supporting evidence by binding analyses such as SPR and/or mutant viruses. The authors have done so for the N392 glycan but it would be good to see similar analyses for the GDIR vs GNIR motifs (see also below), for the N386 glycan and for the V1-residues 138 and 139.

A comprehensive analysis of BG18 neutralization of mutant viruses was presented in Table S4 from Freund et al., 2017, Science Translational Medicine (reproduced below). This includes what the reviewer requested by showing IC₅₀ values in TZM-bl neutralization assays for BG18 against mutant viruses with glycan deletions at N386 and N392. To address the reviewer’s requests for information about other residues, the paper includes an analysis of BG18 neutralization of viruses with GDIR versus GNIR motifs (Table 2), which demonstrates that BG18 is less sensitive to this variation than PGT121 and PGT122. An analysis of variation at the 138 and 139 positions suggested only small effects (Table R1 for reviewers), but these positions are in a highly variable part of V1 (Fig. R1 for reviewers) that is not well-defined in sequence alignments, thus effects of substitutions at these positions would be strain-specific. The revised paper notes that the BG18 contacts with gp120 positions 138 and 139 are likely specific to Envs with V1 characteristics similar to BG505, since similar V1 conformations were not observed in the B41-BG18 structure (Fig. 6 legend and Figure R1 below).

Table R1. BG18 neutralization potency based on glycan and sequence preferences at V1-loop residues 138 and 139 in the presence of N332_{gp120} glycan

IC ₅₀ Values (µg/mL)			
Residue at gp120 position 138	BG18 Potency	Residue at gp120 position 139	BG18 Potency
Asn	0.14 (n=19)	Asn	0.04 (n=26)
Thr	0.06 (n=17)	Thr	0.06 (n=16)
Ser	0.02 (n=15)	Val	0.04 (n=4)
*---	0.05 (n=6)	*---	0.06 (n=8)
+ glycosylation	0.20 (n=12)	+ glycosylation	0.06 (n=20)
- glycosylation	0.05 (n=68)	- glycosylation	0.06 (n=60)

*--- Represents the occurrence of a sequence gap at this position based on HIV-1 strain alignment.

Figure R1 | gp120 V1-loop conformations across various HIV-1 strains. Analysis of V1-loop after alignment against gp120 shows how residues 138 and 139 (sticks) adopt multiple conformations in the presence of V3-glycan targeting bNAbs.

	BG8	BG18	PGT121	12A12
BG505 (T332N)	0.007	0.001	0.012	0.030
N137A	0.003	0.001	0.002	0.027
N156A	0.008	0.001	0.007	0.024
N295A	0.009	0.001	0.014	0.044
N301A	0.004	0.001	0.011	0.027
N339A	0.007	0.001	0.016	ND
N386A	0.004	0.001	0.015	0.030
N392A	0.005	0.001	0.006	0.012
	BG8	BG18	PGT121	12A12
JR-CSF	0.009	0.001	0.011	0.233
N295A	0.006	0.001	0.016	0.203
N301A	0.006	0.001	0.011	0.010
N332A	> 50	> 50	> 50	0.240
N339A	0.010	0.001	0.007	0.028
N362A	0.008	0.001	0.018	0.267
N386A	0.012	0.001	0.024	0.332
N392A	0.014	0.002	0.017	0.230
G324A	0.005	0.001	0.044	0.236
D325A	0.005	0.001	0.079	0.203
I326A	0.005	0.001	0.024	0.278
R327A	0.005	0.001	0.018	0.131
Q328A	0.023	0.001	0.057	0.038
H330A	0.050	0.001	0.017	0.288
	BG8	BG18	PGT121	12A12
92BR020	0.015	0.001	0.004	0.050
N136A	0.005	0.001	0.002	0.027
N156A	0.011	0.001	0.003	0.035
N301A	0.009	0.001	0.003	0.011
N332A	> 50	> 50	0.080	0.039
N386A	0.012	0.003	0.006	0.055
N392A	0.023	0.001	0.007	0.105
D325A	0.006	0.001	0.006	0.050
I326A	0.006	0.001	0.002	0.011
R327A	0.011	0.001	0.001	0.022
Q328A	0.018	0.001	0.002	0.028
H330A	0.018	0.001	0.003	0.073
N136A + N332A	> 50	> 50	> 1	0.025
N156A + N332A	> 50	> 50	> 1	0.023
N301A + N332A	> 50	> 50	> 1	0.007
N136A + N156A + N301A	0.001	0.001	0.001	0.013

Table S4. IC50 and IC80 values in TZM-bl assay of antibodies BG8, BG18, PGT121 and 12A12 against selected HIV-1 V3 envelope mutants

The comparison of BG18 with other V3-base bNAbs could be extended a bit. The comparison between BG18 and 10-1074 in Fig. 5 is great, but why not broaden the comparison on several parameters. The different angles of approach of various V3-base bNAbs are already compared in Fig. S4 and possibly this figure could be extended.

The comparison of BG18 with other V3-base bNAbs was limited to structures in which the bNAb was bound to a trimeric Env, thus we compared the GDIR interactions of BG18 with those of 10-1074 in Figure 5. The 10-1074 interactions with the GDIR and the V1-loop are similar to PGT121-124 interactions as shown in Figure 3 (reproduced below) in Gristick et al, 2016, NSMB:

Thus, comparison of BG18 with PGT121-124 bNAbs would be redundant. While BG505-PGT128 cryo-EM and crystal structures exist (PDBs 5ACO and 5C7K, respectively), their low resolution (>4.4 Å) occludes accurate placement of side-chains involved in potential H-bond or hydrophobic interactions; therefore, a meaningful comparison would not be obtained.

1. The authors could include a neutralization potency/breadth plot for PGT121, PGT128, PGT124, 10-1074, PGT135 and BG18 showing that BG18 is the most potent (assuming the relevant data is available in the existing literature).

Figure 1E in Freund et al., 2017, Science Translational Medicine shows coverage curves for BG18, 10-1074, PGT121, NC37, and BG1. We added a new figure in this paper (Supplemental Fig. 1) to include PGT128, PGT124, and PGT135 coverage curves.

2. The protein contacts (buried surface area per amino acid) could be tabulated for each antibody. These would include the amino acid contacts of the GDIR motif, in the V1-loop, and others if relevant.

The revised paper includes buried surface areas per amino acid for our highest resolution structure and the 3.9 Å natively-glycosylated 10-1074-BG505-IOMA structure (PDB 5T3X) (Table S1), with caveats in the legend explaining that buried surface area calculations can be inaccurate for low resolution structures. Given the different resolutions of existing structures, together with the fact that the Env antigens in these structures differ (some are gp120 monomers; some are trimers; some include only high mannose glycans; some were treated with Endo H, etc.), we think that further direct comparisons between other V3-glycan bNAbs would not yield meaningful conclusions.

3. The glycan contacts (buried surface area per individual glycan) could be tabulated. I assume this would include the glycans at N133, N137, N156, N301, N332, N386, N392, and others if relevant.

Although it is possible to calculate buried surface areas per glycan, not all of the glycans at each position were ordered, so the resulting values would not be meaningful. However, inclusion of BSA for glycans at N156, N332, N386, and N392 are now included in Table S1.

4. The (overlapping) footprints of the different bNAbs could be shown on the trimer structure.

Thank you for this suggestion. We have expanded upon the BG18 and 10-1074 comparison in Fig. 4 by showing analogous footprints for other V3/N332_{gp120}-glycan bNAbs, as well as where they overlap with BG18 in Supplemental Fig. 5.

Despite the low resolution, the B41 structure might be exploited a bit more. The authors observe the difference between the GDIR and GNIR motifs. It would be good to see SPR data on the B41 trimer containing a GDIR motif and the BG505 trimer with a GNIR motif to show the effect of the amino acid substitution on binding affinity.

Please see our response above regarding GDIR versus GNIR. With respect to potential SPR experiments, we have been unable to obtain interpretable SPR sensorgrams for bNAb Fab interactions with B41 SOSIP trimers. We do not understand this, as sensorgrams were interpretable for BG505 SOSIP trimers. Rather than derive possibly inaccurate binding affinities from SPR experiments, which would be in the context of a single strain, we include what we believe is a more meaningful analysis of the effects of GDIR versus GNIR motifs on bNAb neutralization potencies (relevant portion of Table 2; reproduced below). The table reports geometric mean IC₅₀ values (µg/mL) across strains including the N332 glycan (number of strains indicated in parentheses). This analysis suggests that BG18 and 10-1074 are more tolerant of the D → N substitution in the GDIR motif than PGT122 and PGT121.

	BG18	10-1074	PGT122	PGT121
D325	0.05 (n=65)	0.05 (n=209)	0.11 (n=89)	0.07 (n=229)
N325	0.10 (n=12)	0.12 (n=47)	4.82 (n=12)	0.41 (n=49)

If I got this correct, by default, glycans were modeled as Man5-9GlcNAc2 glycans, UNLESS density was clearly interpretable for fucose and/or moieties emanating from the oligosaccharide core. However, there is now a wealth of emerging site-specific glycan composition data from the Paulson lab and in particular the Crispin lab that be exploited here. Instead of modeling Man5-9GlcNAc2 glycans by default, the authors could model the most commonly found glycoform species from the site-specific analysis of the Crispin lab (unless density is clearly interpretable and indicates that a different glycoform is present compared to the most common one in the Crispin analysis).

Because a subset of differentially glycosylated protein may crystallize out of a heterogenous mixture, we can only use the mass spectroscopy analysis of purified protein as a guide for building; therefore, to avoid over-interpretation of our model, we only used interpretable electron density for modeling glycans at potential N-linked glycosylation sites (PNGSs). In our first natively-glycosylated Env structure paper (Gristick et al., 2016, NSMB), we included comprehensive tables (Supplemental Fig. 5-8) that compared glycan forms seen by crystallography, cryo-EM, and by mass spec experiments in the Crispin lab for individual PNGSs. Rather than reproduce the same information in this paper, the revised paper refers to those tables in the Supplemental Fig. 7 legend, from which direct comparisons can be made.

Some glycans are mentioned as part of the V3-base (N386, N392 in abstract and end of introduction; and also “N137, N156 and N301 at the base of V3”, results page 10). This is ofcourse correct for N301, but many virologists would not consider N386 and N392, and N137 and N156 as being located at the V3 base, but in V4 or V1, respectively. I realize that they are close to the V3-base in the 3D structure, but the wording is a bit ambiguous. I would propose to change the wording to express that they are in close proximity of the V3-base.

Done.

The designation “V3-targeting bNAbs” (title and elsewhere) might cause some confusion. Traditionally, V3 epitopes have been regarded as targets for nonneutralizing Abs or NAbs for only unusually sensitive viruses that expose their V3 constitutively. Such Abs usually target the tip of V3. In contrast, bNAbs such as BG18 target the V3-base and surrounding glycans (and even some residues in V1). Rather than V3 antibodies I would suggest terming them V3-base or V3-glycan antibodies.

We have used the term V3/N332 throughout the paper to mean that these are V3-glycan antibodies. The reviewer is correct that the title of the paper was ambiguous – this has been changed from V3-targeting bNAb to V3-glycan bNAb.

Although the structures contain the IOMA and 35O22 bNAbs as well as BG18, they are not further discussed in the paper. I assume that the structures did not reveal anything new on these bNAbs, but if that is the case, perhaps the authors could acknowledge this in one sentence.

BG18 did not alter binding mode at the gp41/gp120 interface or CD4bs by 35O22 or IOMA, respectively, as these Fabs adopted similar conformations as previously observed on trimeric Envs. This is now discussed in one sentence in the Results section.

In the first paragraph of the discussion the authors mention that their new structure represents the second structure with fully and naturally glycosylated Env. I assume this is true for the existing crystal structures, but what about the existing cryo-EM structures? Perhaps that authors can explain/expand.

The reviewer is correct that there are cryo-EM structures of natively-glycosylated Envs, which we compared to our first natively-glycosylated Env crystal structure in Gristick et al., 2016. We have revised the Discussion to clarify that we meant that our structure represents the second natively-glycosylated Env crystal structure and included a citation to a natively-glycosylated cryo-EM structure in the revised introduction.

*Table 2. What does the * stand for?*

This omission has now been corrected in the revised manuscript.

Reviewer #2 (Remarks to the Author):

Barnes et al. report three crystal structures of a potent V3/N332-directed broadly neutralizing antibody (bNAb) BG18 in complex with natively-glycosylated, clade A and clade B SOSIP Env trimers at 3.8Å, 4.9Å and 6.7Å, respectively. A 3.8 Å X-ray free electron laser dataset was necessary to give an electron density map that allowed interpretation of most amino acid side chains, in particular, those at the antibody-Env interface. These structures show a different angle of approach by BG18 to gp120 when compared to other members of the PGT121/10-1074 bNAb family. This unique orientation allowed additional contacts with glycans at N392 and N386 near the V3 base, as well as contacts with the polypeptide chain of the V1, which might explain the increased potency of BG18 in neutralization. The structural information could potentially guide development of V3/N332-directed therapeutic antibodies and immunogen design targeting this important epitope.

The manuscript is clearly written, and data are solid and well presented. It should be of interest to the HIV vaccine community and general readers of Nature Communications. A small concern is their interpretation of glycan density at relatively low resolutions. This is a

very difficult problem because of the intrinsic heterogeneity of glycosylation and the authors have indeed tried hard to “avoid over-interpretation of the electron density”. The question is – how reliable is it to build complex-type glycans simply based on a core fucose? Although core-fucosylation is mainly associated with complex sugars, it has been reported that short high-mannose sugars can also be core-fucosylated.

The Crispin lab analysis of BG505 SOSIP (Behrens et al., 2016, Cell Reports) did not report core fucosylation of any high mannose glycans. Core fucosylation of high mannose glycans has been observed in mutant cell lines (e.g., GnT I-deficient cells) that cannot process high mannose glycans into complex-type glycans, as well as in the presence of 20 μ M swainsonine (a Golgi α -mannosidase II inhibitor) (Lin A, et al., 1994, Glycobiology; Crispin M, et al., 2006, Glycobiology). However, in the latter manuscript, Crispin et al., suggested that “*the paucity of fucosylated oligomannose glycans in glycoproteins expressed in wild-type mammalian cells can, thus, be attributed to such GnT I-dependent pathways which act to deplete both the predominant substrate (Man₅GlcNAc₂) and the product (Man₅GlcNAc₂Fuc) of the GnT I-independent fucosylation pathway.*” Thus, since we were not using a mutant GnT I-deficient cell line or swainsonine during protein expression, core fucosylation should be limited to complex-type glycans.

In addition, as the authors pointed out, it seems that some complex type N-glycans previously modeled at positions N160, N276, N301 and N392 are not supported by their current data sets. It is therefore generally a good practice to interpret only what the map can show. Otherwise, it would be misleading, particularly for those non-structural biologists who would believe everything built in the model.

We agree that it is best to interpret only what electron densities maps show, which is what we did for building the models for the current structures. The submitted paper included a short summary of what was a lengthier discussion of how glycans were built into our first natively-glycosylated Env structure (Gristick et al., 2016, NSMB), which was subsequently fully elaborated upon in a methods paper (Gristick et al., 2017, Acta Cryst D). The Experimental Methods section of the revised paper directs the reader to these papers for more details.

Reviewer #3 (Remarks to the Author):

This MS reports crystal structures of two different SOSIP constructs (a clade A, BG505) and a clade B (B41) bound with the Fab of a recently characterized, V3-directed antibody, called BG18. The structures include BG18 with BG505 and a gp120/gp41 interface Fab (35O22), BG18 with BG505 and IOMA (a CD4bs Fab), and BG18 with B41 and 35O22. The gp140s have native glycosylation, which is relevant to this class of antibodies, which contact both protein and glycan. A data set for the first of these three structures, recorded on the X-FEL at LCLS, extended the resolution of the structure modestly but usefully (from 4.1 Å to 3.8 Å).

Analysis of BG18 bound with gp140 adds to the increasingly extensive portfolio of structures of various bnAbs in a more or less “functional” context. The structure determinations are solid, the descriptions detailed (too detailed for this reader), and the comparisons with PG121 and other V3 antibodies useful. With revision, the MS will be appropriate for publication by Nature Communications.

What do I mean by “with revision”? Four principal points.

(1) The description of the contacts, comparisons, etc., is unnecessarily complicated. As far as I could tell, what matters is that the footprint of this antibody is somewhat different from the footprint of the others, in defined ways (with consequences for the overall pose of the domains) that should be relatively easy to outline. That is all a reader can comprehend,

without having the coordinates, and when one has the coordinates, a brief statement of what to look for is all that's needed.

Sentences describing residue level descriptions of potential H-bond and hydrophobic interactions have been moved to the legends for Figs. 5 and 6.

(2) The authors try to make a case for the advantages of this particular antibody (with respect to other V3 antibodies) -- as a therapeutic, or perhaps as the sort of V3 antibody one might like to elicit with a suitably designed immunogen. The tone throughout smacks a bit of "my daddy can beat up your daddy" -- methinks the authors do protest too much, however, so I come out a skeptic instead. A succinct comparison of pros and cons in the Discussion is fine, but otherwise, just give the readers the facts and let them come to their own conclusions.

We have revised the Discussion to remove most of the comments about potential advantages of BG18 compared with other V3-glycan bNAbs.

(3) The X-FEL data were clearly helpful. But the comparison isn't a fair one, as the characteristics of the x-ray beam they used at SSRL were of course different, not just in brightness but also in collimation, diameter, etc. In other words, there are advantages to a very small, well-collimated, very bright beam if your crystal is small (these weren't SO small), but perhaps GMCA-CAT at APS would have done as well (with a 10 micron beam, helical scanning or similar strategy, etc.) A comparison isn't the purpose of this report, of course, so I believe that the authors should remove "as revealed by an X-ray free electron laser" from the title. Yes, given the resources these authors had to hand, that's how they did it, and they obviously did a great job. But the hidden implication is that you need an X-FEL rather than a better synchrotron beam. This reviewer is unconvinced.

The reviewer is correct that we don't actually know how much, if any, the XFEL source improved the data. We've therefore changed the title.

(4) "Angle of approach" is a poor term (pace Andrew Ward). For several reasons. First, there's a good, much better, term used generally in chemical biology -- "pose". (Note: only four letters!). Second, what really matters in the case of antibodies is the footprint and its details, not the "angle" (badly defined in any case, as "angle" is a 2D term; in 3D, there are at minimum two parameters, e.g., direction cosines of an axis, and one needs a common reference axis in the two structures as well).

Although we've used the term "angle of approach" in some of our previous papers, we agree with the reviewer that the term is misleading and not precise. We have modified the text accordingly.

There is also a point that I'd like to see addressed, although the resolution of one of the structures limits what what one can say. Nonetheless, I'd like to know whether the IOMA bound gpp140 is in any way different in conformation from the 35O22 bound gp140 (BG505 in both cases). The BG12 Fab may have an influence, potentially minimizing any conformational difference induced by IOMA, but a comment would be useful.

Because our structure with IOMA is only 6.7 Å resolution, a better comparison to address this question might be to align the gp140 in the 3.5 Å BG505-IOMA-10-1074 structure (Gristick et al, 2016) with one of the BG505-35022-PGT121 family structures (e.g., 5CEZ at 3.0 Å). (This comparison is reasonable since 10-1074 and members of the PGT121 family bind in similar orientations.) Such an analysis shows a 0.8 rmsd when aligning against 446 C α atoms of gp120 in

the 10-1074-BG505-IOMA complex with the PGT121-BG505-35O22 complex, with the largest deviations occurring near the CD4bs due to IOMA binding.

A few trivial items:

lines 30-32 Our structures show that common, D-gene encoded residues have conserved contacts with the N332 glycan and with the GDIR peptide motif, even though the pose of the Fab is different from those of V3/N332 bnAbs in the PGT121/10-1074 family. (or, to save words, "from those of PGT121/10-1074-family V3/N332 bnAbs"). [I believe we should reserve "reveal" for religious revelation; "show" is quite good enough for down-to-earth science. I concede that "reveal" is OK in sentences like "Einstein's theories revealed a deep connection between space and time", but that was a general, epistemological revelation, not a simple statement of observation. Note also active voice in subordinate clause.]

line 198 contrasts with interactions (contrasts doesn't take a definite object)

lines 206-207 N137 and N156 are NOT at the base of the V3 loop -- reword the sentence

line 232 contrasts with

line 236 misuse of "rationalize that". These differences in light chain interactions reflect the germline origins of the BG18 and PGTxxx light chains, which derive from different VL gene segments.

line 241 ...GNIR sequence instead of GDIR, allowing us to analyze the D325N substitution... (current version ungrammatical)

line 245 ...GNIR motif analogously to how it recognizes GDIR.

lines 254 and 260 delete "surprisingly" Let the reader decide whether it is surprising or not.

We have revised the manuscript based on these suggestions and the other comments from the reviewers and thank them for careful reading of the paper. We hope the revised paper is suitable for publication in Nature Communications.

REVIEWERS' COMMENTS:

Reviewer #1 (Remarks to the Author):

I would have wished to see some SPR analyses to solidify the role of specific B18 contact residues (the authors do acknowledge that they attempted to perform such studies with b41 and did not yet succeed), but otherwise the authors have addressed my concerns satisfactorily.